# Investigating firn structure and density in the accumulation area of the Grosser Aletschgletscher using Ground Penetrating Radar

Akash M Patil<sup>1,2</sup>, Christoph Mayer<sup>2</sup>, Thorsten Seehaus<sup>1</sup>, Alexander R. Groos<sup>1</sup>, and Andreas Bauder<sup>3,4</sup>

Correspondence: Akash M Patil (akash.patil@badw.de / akash.patil@fau.de)

Abstract. The role of firn structure and density in geodetic glacier mass balance estimation has been constrained, with studies in alpine conditions primarily relying on models. Our research focuses on understanding the firn structures, density, and accumulation history in the Grosser Aletschgletscher accumulation area, using field methods mainly involving Ground-Penetrating Radar (GPR) as a geophysical tool, and glaciological methods such as snow pits, snow cores, firn cores, and isotope analysis. We characterise the firn structure and determine the spatial firn density-depth profiles by estimating electromagnetic wave velocities using the GPR-based Common Mid-Point (CMP) method. This is done by identifying reflection hyperbolae using semblance analysis of the CMP data set. Three density-depth profiles, up to 37 m depth, were obtained at various locations within the glacier accumulation area. The Ligtenberg (LIG) and Kuipers Munnekee (KM) firn compaction models were selected from the Community Firn Models (CFM) to evaluate how well the model results matched the observations. These models were predominantly adjusted to fit the estimated 1-D firn density profiles from CMP measurements by optimizing model coefficients based on regional Alpine climatic conditions, rather than the conventional method of tuning to the firn core density profiles. Further, a method is introduced to estimate accumulation history by chronologically identifying GPR-derived Internal Reflection Horizons (IRHs) as annual firn layers, by comparing the estimated Snow Water Equivalent (SWE) within each IRH to SWE from long-term point mass balance measurements available at the accumulation area of the glacier. We investigated the spatial distribution of the firn density and the glacier's accumulation history over the past 10-14 years (2010-2023) using a 1.8 km GPR transect, supported by CMP-derived density-depth profiles. Furthermore, our findings emphasize the importance of direct measurements, such as snow cores, firn cores, and isotope samples, in identifying the previous end-of-summer horizon. In this study, we demonstrate the potential of integrating GPR, direct measurements, and firn compaction models to monitor firn structures and density, ultimately enhancing glacier mass balance estimation in future research.

## 20 1 Introduction

Mountain glaciers connote the impact of climate change along with their local hydrology and ecological importance (Haeberli, 1998; Kaser et al., 2006; Stocker et al., 2013). Glaciers around the world are monitored by studying their mass balance to

<sup>&</sup>lt;sup>1</sup>Institute of Geography, Friedrich-Alexander-Universität Erlangen-Nürnberg, 91508 Erlangen, Germany

<sup>&</sup>lt;sup>2</sup> Bavarian Academy of Sciences and Humanities, Geodesy and Glaciology, Alfons-Goppel Str. 11, D-80539 Munich, Germany

<sup>&</sup>lt;sup>3</sup>Laboratory of Hydraulics, Hydrology and Glaciology (VAW), ETH Zurich, Zurich, Switzerland

<sup>&</sup>lt;sup>4</sup>Swiss Federal Institute for Forest, Snow and Landscape Research (WSL), Sion, Switzerland

understand the response and adjustment to climate change and provide information on water sources (Barry, 2006; Kaser et al., 2006; Ohmura et al., 2007; Zemp et al., 2009). Traditional mass balance measurements are susceptible to higher uncertainties than the geodetically estimated volume change (Sold et al., 2015). Therefore, geodetic approaches are an important and efficient way to estimate glacier volume change, predominantly in Alpine glaciers and glacierized mountain regions (Sapiano et al., 1998; Cuffey and Paterson, 2010; Huss et al., 2014). However, there is a need to convert this volume change to mass change. One approach involves assuming a constant conversion factor  $850\pm60~{\rm kg~m^{-3}}$  (Huss, 2013), considering the varying amounts of snow and firn relative to the glacier size. Another approach is contemplating constant densities each for the glacier's accumulation and ablation area (600 kg m<sup>-3</sup> in the firn zone and 900 kg m<sup>-3</sup> in the ablation zone) to derive mass balance from zonal surface elevation changes (e.g. Schiefer et al., 2007; Moholdt et al., 2010; Kääb et al., 2012; Bolch et al., 2013). To improve the accuracy of glacier mass balance estimations, contemporary research primarily focuses on reducing uncertainties in conventional mass balance measurements and geodetically derived volume changes (Huss et al., 2009; Fischer, 2011; Zemp et al., 2013).

As exemplified in many geodetic studies, uncertainties in converting glacier volume to mass change by assuming a constant density value are susceptible to inaccuracy in the accumulation area, where the firn densification processes are poorly constrained (e.g. Gardelle et al., 2012; Kronenberg et al., 2016). Even with density assumptions, uncertainties in the mean glacier mass balance can arise from long-term firn compaction. Processes such as overburden pressure from accumulated fresh snow and the formation of ice lenses due to melting and refreezing within the firn body alter the firn stratigraphy, reflecting changes in climatic conditions (Benson, 1996). In addition, Jordan et al. (2008) illustrates that the percolation and refreezing of meltwater result in spatial alteration of snow and firn structures. Earlier research provided the fundamental understanding of the firm densification processes in sub-zero climatic conditions (e.g. Herron and Langway, 1980; Arthern and Wingham, 1998; Li and Zwally, 2004; Reeh, 2008a; Ligtenberg et al., 2011), but not many studies have looked at the firn densification in Alpine glaciers where the rate of firn densification is significantly higher than in cold firn (Kawashima and Yamada, 1997; Cuffey and Paterson, 2010). Firn densification on mountain glaciers can be influenced by the presence of refrozen meltwater within the pore spaces (Schneider and Jansson, 2004) and the cooling of the uppermost firn layers during the winter season (Hooke et al., 1983). Similarly, comprehension of the firm density structure has many glaciological applications, such as estimating glacier mass changes (Shepherd et al., 2012), firn core studies to determine its age (Bender et al., 1997; Blunier and Schwander, 2000), and the contribution of surface meltwater to the glacier hydrology in the case of mountain glaciers (Stevens et al., 2024). Whereas, studies like Huss (2013) emphasise the detailed understanding of spatial firn volume and density distribution to reduce uncertainties in accurately estimating mountain glacier mass balance from measured volume using geodetic methods. Thus, understanding the firn stratigraphy and density in the accumulation area is highly important to comprehend the processes influencing firn structure and densification.

Previous studies have shown that glaciological methods such as snow pits represent pragmatic and comprehensive annual accumulation conditions (Mayer et al., 2014), and firn cores provide information on ice layer patterns resulting from accumulation and surface melt (Machguth et al., 2016; Rennermalm et al., 2022). Findings from various studies (e.g. Parry et al., 2007; Dunse et al., 2008; Brown et al., 2011; Marchenko et al., 2017; Heilig et al., 2020) have revealed local variability of snow depths,

firn densification rates, and long-term accumulation rates. However, each method has pros and cons concerning the effort and vertical resolution required to analyse the stratigraphy. Methods like isotope sample analysis better allow the identification of the annual layers (Aizen et al., 2005) within the firn body. Still, they are constrained to the colder climate owing to the effect of melting processes in temperate glacier conditions, resulting in the dispersion of isotope samples (Hou and Qin, 2002). The main drawback of these conventional glaciological studies is the time required for excavating the snowpit or firncore sample, which non-invasive geophysical methods can compensate for.

Ground-penetrating radar (GPR) has been effectively used in various glaciological applications, such as snow-accumulation studies in Antarctica (Sinisalo et al., 2003), firn and ice transition in a polythermal glacier in Syalbard (Pälli et al., 2003), and estimation of firn density in the percolation zone of the western Greenland ice sheet (Brown et al., 2012). Additionally, it has been utilized to study firn stratigraphy in Svalbard (Marchenko et al., 2017) and for accumulation and thickness measurements of Alpine and high-altitude glaciers (e.g. Machguth et al., 2006; Bauder et al., 2018; Lambrecht et al., 2020). The GPR's ability to detect the change in dielectric permittivity and electric conductivity between two mediums, the ease with which it can be used, and the possibility of having repeat measurements due to its non-invasive behaviour (Davis and Annan, 1989; Fisher et al., 1992; Arcone and Kreutz, 2009) make it the preferred technique over other geophysical methods. Radar-based subsurface mapping can reveal the spatial variability of accumulation rates (Waddington et al., 2007; Eisen et al., 2008). According to Vaughan et al. (1999) and Helm et al. (2007), the visualised internal reflection horizons (IRHs) within the firn body obtained from the GPR are perceived to be isochrones. Several studies in polar and sub-polar regions (e.g. Wadham et al., 2006) demonstrated the identification of IRHs as a previous end-of-summer surface to quantify the annual accumulation rates. Various studies (e.g. Kohler et al., 1997; Kruetzmann et al., 2011; van Pelt et al., 2014) have shown the ability to determine the water equivalent (w.e.) between the IRHs using radar data in cold conditions, but not in temperate mountain conditions. However, Sold et al. (2015) showed the usefulness of helicopter-based GPR measurements on a temperate valley glacier (Findelengletscher) to unlock the firn annual water equivalent in temperate conditions. Nevertheless, the study does not consider the importance of radar velocity to get the IRHs depths, but rather depends on simple firn densification models and firn core density profiles. As Sold et al. (2015) study was conducted on temperate firn, where the possibility of high melt in temperate conditions results in the presence of liquid water due to significant melt before or during the data acquisition. This reduces the radar wave propagation velocity, resulting in overestimation of density (Bradford et al., 2009). Therefore, the application of the GPR-based CMP method in temperate firm is prone to higher inaccuracy, which is discussed further in our study. The latest study by Bannwart et al. (2024) uses the GPR on the Grosser Aletschgletscher to compare the elevation bias and the impact of snow layers on radar signal propagation, along with point snow measurements and spaceborne radar.

The GPR-based Common Mid-Point (CMP) measurements are typically used to determine the radar propagation velocity as a function of depth, which provides a more accurate estimation of IRH depths observed on radargrams. Hempel et al. (2000) and Brown et al. (2011) used the GPR-based CMP survey to estimate firm column density variations with depth in the percolation zone of the Greenland Ice Sheet (GrIS). Whereas Booth et al. (2013) shows the application of the GPR-derived CMP method in a polythermal mountain glacier (Storglaciären) in Sweden, discussing the benefits of CMP gather measurements in velocity and density estimations. Booth et al. (2013) study, further reiterating the efficiency of GPR as a geophysical tool to

complement mass-balance measurements. So, we observe the application of the GRP-based CMP method in polar conditions; however, to our knowledge, there are no studies in Alpine conditions focusing on firn density-depth profiles using GPR-derived CMP gather.

In this study, we explore the firn structure and density distribution using the GPR-based CMP method, alongside established glaciological methods, in the accumulation area of the Grosser Aletschgletscher in the Swiss Alps. We identified IRHs on radargrams as annual firn layers by comparing CMP-derived Snow Water Equivalent (SWE) with SWE from long-term point measurements available from GLAMOS (2024), which helps in tracking the accumulation history over the last 14 years. We also obtained a long GPR transect to monitor the spatial distribution of the firn layering. Tracking the depth to the last summer horizon using glaciological measurements is crucial for validating GPR-derived results. The geophysical results presented here lay the foundation for testing and calibrating firn compaction models selected from the Community Firn Model (CFM). By doing so, we aim to evaluate their accuracy under specific climatic conditions and their usefulness in Alpine conditions for future research. However, this study primarily outlines the understanding of firn structure and density using field observations. Consequently, we illustrate the application of the GPR and highlight the significance of the CMP method in investigating the spatial variation in firn stratigraphy and density structure, as well as in retrieving the accumulation history of the firn area, exemplified at the Grosser Aletschgletscher.

## 2 Study area and Data acquisition

# 2.1 Study area

100

In search of an Alpine glacier with a thick and extensive firm body, we selected the Grosser Aletschgletscher in the Swiss Alps as a study area for this research, which has good accessibility from the High Altitude Research Station Jungfraujoch. With an area of about 80 km<sup>2</sup>, a length of about 20 km, and a maximum thickness of around 800 m, the Grosser Aletschgletscher is the biggest glacier in the European Alps (Wadham et al., 2006; Farinotti et al., 2009; GLAMOS, 2018). The Grosser Aletschgletscher ranges from 1700 to 4200 m a.s.l. and comprises about 20% of the total glacier volume in the Swiss Alps (Bauder and others, 2007). Accumulation areas of the glacier consist of Grosser Aletschfirn in the west, Jungfraufirn in the 115 north, and Ewigschneefeld in the northeast (Fig. 1). According to meteorological data recorded at Jungfraujoch research station at an elevation of 3580 m a.s.l., the mean annual temperature of the last three decades (1991-2020) was -6.7 °C, and the mean daily temperature from the September-June period was well below 0 °C (MeteoSchweiz, 2024). The long-term point measurement is located at the accumulation area of the glacier at an elevation of 3390 m a.s.l., near the Jungfraufirn (denoted as "Stake" in Fig. 1). It was installed in 1918 and has presented continuous annual glaciological measurement records since 120 then (GLAMOS, 2018). The glacier has been extensively studied since the 1940s (e.g. Seligman, 1941), and it provides an archived glaciological and other in situ datasets as available in GLAMOS (2024) and the World Glacier Monitoring Service (WGMS). We present Table 1, which summarizes the different observations presented in this study.

Figure 1. The study area with the geophysical and glaciological measurements on two of the accumulation basins of the Grosser Aletschgletscher glacier. Snow pits (SP1), a firn core (FC1), and snow cores (SC1 and SC2) were obtained from the Ewigschneefeld and Mönchsjoch along with the GPR profile during the February-March 2024 measurements. The three Common Mid Point measurements (CMP2, CMP3, and CMP4) represented by red cross (pointed by black arrows) and shallow snow pits (SP2, SP3, and SP4) were gathered during 16-17 May 2024 at Mönchsjoch, Ewigschneefeld, and Jungfraufirn (near the long-term in-situ/point mass-balance measurement, which is represented as Stake in this figure), respectively. Because of weather and time constrain, CMP at the far end of the long GPR profile (near SP1) was not possible. The inset map illustrates an oblique satellite view of the Grosser Aletschgletscher glacier in winter 2024 (© Google satellite map). The background map is a cloud-optimised GeoTIFF of 2 m resolution provided by the Swiss Federal Office of Topology (© swisstopo 2024).

**Table 1.** Summary of the different glaciological and geophysical observations acquired during two expeditions in winter 2024 as shown in Fig. 1. All observations are sorted based on the acquisition date.

| Observations type | Survey date      | Latitude             | longitude          | Elevation (m a.s.l) |
|-------------------|------------------|----------------------|--------------------|---------------------|
| Snow pit (SP1)    | 29 February 2024 | 46.55155             | 8.02771            | 3377                |
| Snow core (SC1)   | 29 February 2024 | 46.55155             | 46.55155 8.02771   |                     |
| Firn core (FC1)   | 29 February 2024 | 46.55155             | 46.55155 8.02771   |                     |
| GPR transect      | 29 February 2024 | 46.55685 to 46.55155 | 8.00982 to 8.02771 | 3553 to 3377        |
| Snow core (SC2)   | 2 March 2024     | 46.55198             | 8.00548            | 3604                |
| CMP2              | 16 May 2024      | 46.55198             | 8.00548            | 3604                |
| Snow pit (SP2)    | 16 May 2024      | 46.55198             | 8.00548            | 3600                |
| CMP3              | 17 May 2024      | 46.55198             | 8.01411            | 3443                |
| CMP4              | 17 May 2024      | 46.54322             | 7.98396            | 3340                |
| Snow pit (SP3)    | 17 May 2024      | 46.55198             | 8.01411            | 3470                |
| Snow pit (SP4)    | 17 May 2024      | 46.54322             | 7.98396            | 3340                |
|                   |                  |                      |                    |                     |

# 2.2 Data acquisition

## 125 2.2.1 GPR data

On 29 February 2024, a 1.8 km long GPR transect was measured on the Ewigschneefeld (Fig. 1) using the IDS monostatic 200 and 600 MHz dual-frequency, shielded GPR system. The positioning of the GPR transect was based on a hand-held GPS track. The choice for the location of the GPR profile was to repeat the measurements done by Bannwart et al. (2024) for future studies. Three sets of crossed CMP profiles at a right angle were gathered at three different locations of the upper Grosser Aletschgletscher, i.e., at the Jungfraufirn (near the Stake in Fig. 1), the Mönchsjoch plateau, and the Ewigschneefeld on 16-17 May 2024. The CMP measurements were carried out using the PulseEkko bi-static 500 MHz GPR system, consisting of a separate shielded transmitter and receiver antenna. At each spacing of the GPR antennas, radar pulses were triggered manually twice, with 128 stacks per trace. The initial distance between the two antennas was 20 cm. The transmitter and receiver were moved along a straight line, maintaining a constant offset of 20 cm over a length of 20 m, with a step size of 10 cm on either side of a common midpoint. However, this study includes only an analysis of CMP profiles measured parallel to the ice flow from each location. Table 2 shows the general GPR settings used for the data acquisition.

## 2.2.2 Glaciological investigations

A 4 m deep snow pit was dug at the Ewigschneefeld on 29 February 2024. The snow density was measured using a 17 cm long metallic cylinder of approximately 4.75 cm in diameter, which resulted in a vertical resolution of 17 cm. A shallow firn core of 3.8 m deep from the bottom of the snow pit was drilled to a total depth of 7.8 m using a "Mark II Ice Coring System" from Kovacs Ice Drilling Equipment with a diameter of 9 cm. The density was measured at a 20 cm depth interval on the

Table 2. The GPR antenna settings used for the GPR profile and CMP measurements during the winter campaign 2024.

| Acquisition date        | 29 February 2024 | 16-17 May 2024 |  |
|-------------------------|------------------|----------------|--|
| Type of acquisition     | GPR profile      | CMP gather     |  |
| Antenna frequency (MHz) | 200 and 600      | 500            |  |
| Number of samples       | 2048             | 500            |  |
| Time window (ns)        | 400              | 400            |  |
| Trace increment (m)     | 0.0176           | 0.2            |  |
| Time increment (s)      | 0.195            | 0.2            |  |

core. Another shallow snow core of 5.8 m was obtained a few meters away from the snow pit, to allow a direct comparison between the snow/firn core and snowpit measurements (Fig. 1). Another snow core of 5.4 m long was recovered from the Mönchsjoch plateau on 2 March 2024, applying the same vertical resolution for the density measurements. In addition, isotope samples were collected every 20 cm from the snow and firn cores and the snow pit at both locations to identify the depth of the end-of-summer horizon. Similarly, snow pits were dug at the Mönchsjoch plateau (1.3 m deep), Ewigschneefeld (1.9 m deep), and Jungfraufirn (1.9 m deep) on 16 and 17 May 2024 (Fig. 1). We used estimated SWE from a point mass-balance measurement located at the accumulation area of the glacier provided by GLAMOS (2024), which is similar to our snow/firn core data. The point mass balance measurement is a well-established reference horizon that helps to compare our accumulation history results from geophysical measurements. Here we differentiated snow and firn core terminology based on available core depths (Bannwart et al., 2024), as the snow core does not reach the firn depth, whereas the firn core reaches the previous year's firn layer.

#### 3 Methods

## 3.1 GPR data processing

The GPR data acquired during the field campaign were processed using the ReflexW software (Sandmeier, 2010), following a traditional processing sequence (Ulriksen, 1982; Annan, 1993; Fisher et al., 1996). We applied a sequence of filters and gains to remove apparent noise and improve the visibility of the IRHs on the radargrams, such as moving the start time to get the first arrival at the surface, applying a de-wow filter, stacking, and using a bandpass Butterworth filter to increase the signal-to-noise ratio, as well as background removal and static corrections to reduce system-induced irregularities (Fig. 2). The IRHs seen on radargrams were picked using the built-in, semi-automatic phase follower tool. The radar propagation velocity was assumed to be constant at 0.21 m ns<sup>-1</sup> (Looyenga, 1965) to visualize the radar penetration depth within the firn body (Fig. 2, right axis).

**Figure 2.** The processed radargram obtained from the Ground Penetrating Radar (GPR) long transect at Ewigschneefeld on 29 February 2024 (blue line in Fig. 1). The GPR profile runs from higher (3553 m a.s.l on left) to lower elevation (3377 m a.s.l on right). Yellow arrows indicate the artefacts from stops during the measurements. The red lines indicate prominent IRHs and potential annual layers at different depths. The left y-axis shows the radar penetration in two-way traveltime (twt) in nano-seconds (ns), and the right y-axis is the depth converted in meters by assuming the radar propagation velocity in firn as 0.21 m ns<sup>-1</sup>. The top x-axis is the distance of the GPR transect in meters.

## 3.2 GPR CMP semblance analysis

The physical properties of the subsurface are derived from the radar wave propagation velocity, facilitating the estimation of reflector depth by converting the travel time to depth in the radargram through the dielectric mixing model (Looyenga, 1965; Topp et al., 1980; Endres et al., 2009). The common-midpoint (CMP) method is an acquisition approach keeping an identical distance between the transmitter (Tx) and the profile centre, as well as the receiver (Rx) and the profile centre at all positions along the acquisition. Radar velocities are estimated by matching the curvature of diffraction reflections from the subsurface targets, assuming horizontal homogeneous layers (Yilmaz, 2001; Annan, 2005; Porsani and Sauck, 2007; Schmelzbach et al., 2012). The CMP data were analysed using the semblance analysis, which measures the coherence of energy between waveforms within analysis windows centred on hyperbolic trajectories (Sheriff and Geldart, 1999).

To estimate the depth of IRHs, the GPR-derived CMP profiles were analysed using the CMP (1D) analysis module within ReflexW software. The initial processing of gathered CMP data was done according to the processing of the GPR profile (Sect.

**Figure 3.** The processed GPR CMP data gathered at Ewigschneefeld on 17 May 2024 (CMP2 in Fig.1). The 20 m long CMP profile depicts various hyperbolic reflections, visible down to a depth of 30-32 m with an assumed constant velocity for two-way traveltime to depth conversion. The left y-axis shows the two-way travel time (twt) in ns, while the right y-axis indicates an approximate depth, assuming a constant radar velocity in firn of 0.21 m ns<sup>-1</sup>. The direct wave (black dotted line) and identified hyperbolic reflectors (solid red lines) are indicated at different depths.

3.1). The linear approximation of the direct wave with an assumed wave velocity of 0.3 m ns<sup>-1</sup> is shifted to zero offsets at its starting point for the static time correction. The semblance analysis was run after setting up the input parameters within the interactive adaptation module and the un-normalized correlation section. The correlation histogram can be seen in the center of Fig. 4. The best fitting velocity, known as root-mean-square velocity (Vrms), can be chosen from the correlation histogram (Fig. 4b and c) by comparing the calculated hyperbolas with the measured hyperbolic reflections. This process continues for all identified layers (red lines in Fig. 4a). Figures A1 and A2 in the appendix show the enlarged version of processed CMP data, which illustrates that IRHs can also be detected at greater depths, allowing us to infer the corresponding velocities. The detailed description of the semblance analysis procedure can be found in Sandmeier's geophysical research-ReflexW user guide (Sandmeier, 2010). Semblance analysis gives selected Vrms velocities (Fig. 4b and c) and corresponding twt of the respective

**Figure 4.** The figure illustrates the semblance analysis of the Common Mid-Point (CMP) data gathered at the Ewigschneefeld (CMP3 in Fig. 1). The ReflexW software was used for the Semblance analysis, which presents the picked hyperbolic pattern or IRHs (a) obtained from the CMP gathers matching the energy coherence (b) to attain picked Vrms velocities corresponding to the semblance pick. The 1-D modelled interval velocity for the picked Vrms can be seen on the right part (c) of the figure. The left y-axis in the figure represents the radar wave two-way travel time in ns. The right y-axis of the figure (a and b) indicates an approximate depth, assuming a constant radar velocity in firn of 0.21 m ns<sup>-1</sup>. In Figure C, the left y-axis shows the depth adjusted to the modelled interval velocity. The X-axis in Figure (a) is the distance in meters, and in Figures (b) and (c) is the pickable Vrms velocity for the corresponding semblance energy.

hyperbolae. Interval velocities are estimated using the Dix Eq. (1) as

$$v_{int} = \sqrt{\frac{v_{\text{rms}_i}^2 twt_i - v_{\text{rms}_{i-1}}^2 twt_{i-1}}{twt_i - twt_{i-1}}},$$
(1)

where,  $v_{\rm int}$  is the interval velocity through layer i,  $twt_i$ ,  $twt_{i-1}$ ,  $v_{{\rm rms}_i}$  and  $v_{{\rm rms}_{i-1}}$  are two-way travel times and root-mean-square (Vrms) velocities through layer ith, and i-1th interfaces, respectively. Here, we can derive interval velocity within firn layers by substituting the picked Vrms and twt from the semblance analysis as suggested by Booth et al. (2011), which then allows to determine the firn physical properties by dielectrical mixing model (e.g. Looyenga, 1965; Topp et al., 1980; Endres et al., 2009). Figure A3 in the Appendix illustrates the estimated interval velocities for picked Vrms from all three CMP data sets. The sensitivity analysis of Vrms picking from CMP analysis and its influence on the interval velocity and density profiles is discussed in Section 5.3.

The corresponding density-depth profile can be obtained by adapting the complex refractivity index method CRIM (Wharton

et al., 1980; Knight et al., 2004) as

$$\rho = \frac{\left(\frac{V_{\text{air}}}{V_{\text{fim}}} - 1\right)}{\left(\frac{V_{\text{air}}}{V_{\text{ice}}} - 1\right)} \rho_{\text{ice}},\tag{2}$$

by using the interval velocity-depth profile estimated from picked Vrms and twt from semblance analysis of the CMP data, and using the Dix Eq. (1). The Eq. (2) illustrates the dependency of the firn density estimations on radar-wave velocity. Here,  $V_{\rm air}$ ,  $V_{\rm ice}$  are radar wave propagation velocities in air and ice (0.3 and 0.17 m s<sup>-1</sup> respectively), whereas the velocity of the radar wave within the firn layer ( $V_{\rm firn}$ ) was estimated from the Dix equation using picked Vrms from CMP semblance analysis and  $\rho_{\rm ice}$  assumed as 920 kg m<sup>-3</sup>.

# 3.3 Firn densification modelling

Firn densification models help convert measured glacier volume change to mass change and understand the rheological properties and firn density-depth profiles within mountain glaciers and ice sheets consisting of a thicker firn body (Gagliardini and Meyssonnier, 1997; Lüthi and Funk, 2000). This study primarily examines observation-derived results to investigate firn structure and densification in temperate conditions. By testing and calibrating firn densification models, we aim to evaluate their accuracy under specific climatic conditions and their usefulness for future research.

The Community Firn Model (CFM), developed by Stevens et al. (2020), is used in this study to model firn densification using regional climate data. The CFM is an open-source firn model framework that can run 13 previously published models. It allows users to choose the particular modular run for the required physical processes to obtain results for the selected module. More details about the CFM workflow can be found in Stevens et al. (2020). We used the Ligtenberg (LIG) and Kuiper Munneke (KM) models, developed to simulate one-dimensional firn densification, considering meltwater percolation and refreezing processes, mainly in Antarctica and Greenland, respectively (Stevens et al., 2020). Generally, both models require accumulation rate and air temperature as input parameters and consider potential melting and refreezing processes in the simulations. The LIG is an empirical approach based on observations from ice cores, providing the general firn densification trend (Ligtenberg et al., 2011). The LIG and KM models are based on Arthern et al. (2010) and differ only in their coefficients, representing different sensitivities of densification processes, as shown in Eqs.3 and 4.

We adopted the same parameters for tuning (Table 3), to fit the observed density-depth profile from glaciological and geophysical methods. Parameter tuning was done by iteratively choosing the best coefficient that fits the GPR-derived CMP and glaciological observed density-depth profiles. The models were run for two elevations, 3600 and 3400 m a.s.l., representing the approximate elevations of Mönchsjoch and Ewigschneefeld, to consider air temperature differences between both locations while keeping the precipitation rate constant. The model spin-up period was set for 100 years, during which the models ran over 10 years (2005-2015), forcing data repeatedly to reach a steady state. After this period, the models used daily input data for simulating the firn density evolution under an assumed constant surface snow density of 300 kg m<sup>-3</sup>.

The general formulation of firn densification in the chosen models assumes a logarithmic relationship with respect to the accumulation rate and an exponential relationship concerning the inverse temperature. The coefficients differ for snow densification

(density  $< 550 \text{ kg m}^{-3}$ ) and firn densification (density  $> 550 \text{ kg m}^{-3}$ ). The general form of this relationship is

$$225 \quad C_0 = \left(\alpha_1 - \beta_1 \ln(\dot{b})\right) \left(0.07 \dot{b} g \exp\left(-\frac{E_c}{RT} + \frac{E_g}{RT_m}\right)\right) \quad \text{for } \rho \le 550 \text{ kg m}^{-3}, \tag{3}$$

$$C_1 = \left(\alpha_2 - \beta_2 \ln(\dot{b})\right) \left(0.03\dot{b}g \exp\left(-\frac{E_c}{RT} + \frac{E_g}{RT_m}\right)\right) \quad \text{for } \rho \ge 550 \text{ kg m}^{-3},\tag{4}$$

where,  $(\dot{b})$  is the accumulation rate in kg m<sup>-2</sup> a<sup>-1</sup>, g is the acceleration due to gravity (9.80 m s<sup>-2</sup>), while  $E_c$  and  $E_g$  represent the activation energy for diffusion and grain growth respectively (60 and 42.4 kJ mol<sup>-1</sup>).  $T_m$  represents the mean surface temperature in Kelvin as described in Ligtenberg et al. (2011) and Simonsen et al. (2013). The coefficients  $\alpha_{1,2}$ ,  $\beta_{1,2}$ , and  $\gamma_{1,2}$  are as in the Table 3 for the two models and our optimization.

Table 3. The LIG and KM model parameter coefficients (Eqs. 3 and 4) and the tuned values to fit the observational results.

| Coefficients | LIG   | KM     | Our values at 3400 m a.s.l |       | Our values at 3600 m a.s.l |       |
|--------------|-------|--------|----------------------------|-------|----------------------------|-------|
|              |       |        | LIG                        | KM    | LIG                        | KM    |
| $\alpha_1$   | 1.435 | 1.042  | 1.445                      | 1.045 | 1.455                      | 1.045 |
| $lpha_2$     | 2.366 | 1.734  | 2.397                      | 1.775 | 2.40                       | 1.765 |
| $eta_1$      | 0.151 | 0.0916 | 0.089                      | 0.006 | 0.125                      | 0.006 |
| $eta_2$      | 0.293 | 0.2039 | 0.505                      | 0.425 | 0.315                      | 0.435 |

# 3.4 Input parameters for firn densification model

## 3.4.1 Climate forcings data

The necessary climate forcings for the CFM runs, such as the daily mean temperature, were obtained from the Jungfraujoch weather station at 3580 m a.s.l. (MeteoSchweiz, 2024). Due to high winds and snow drift, precipitation data is not available from the Jungfraujoch research station. Therefore, daily mean precipitation data were used from observations at the Grimsel research station, situated at 1952 m a.s.l., approximately 25 km northeast of our study area. The precipitation data were scaled to the Jungfraujoch elevation using a simple statistical approach (Sect. 3.4.2). Air temperature data were adjusted to the respective elevation using a constant lapse rate of 6.5 °C km<sup>-1</sup>.

We estimated the seasonal snow melt rate known as Degree Day Factor for snow (DDF snow) by dividing the sum of SWE during each year's summer mass balance measurement period by the sum of positive daily mean temperatures during the same period. We neglected the winter melt, as the mean daily temperature from the September-June period over the last two decades was well below 0 °C (MeteoSchweiz, 2022). Figure 5 shows the estimated DDF snow values over the last seven years, resulting in a mean DDF snow value of approximately 3.38 mm °C<sup>-1</sup> day<sup>-1</sup> and a maximum of 6.61 mm °C<sup>-1</sup> day<sup>-1</sup> during the peak melting period of 2022. This DDF snow value falls within the general assumption of 2.7-11.6 mm °C<sup>-1</sup> day<sup>-1</sup> according to Hock (2003). As most ice sheet modelling studies use a DDF snow value of 3 mm °C<sup>-1</sup> day<sup>-1</sup> (e.g. Wake and Marshall,

2015), we tested the model's behaviour for a DDF snow value of 3.5 mm  $^{\circ}$ C<sup>-1</sup> day<sup>-1</sup>. Furthermore, the chosen CFM model's sensitivity tests were also run for two DDF snow values of 5.5 and 8.5 mm  $^{\circ}$ C<sup>-1</sup> day<sup>-1</sup>.

**Figure 5.** Bar plot showing summer balance and the corresponding Degree Day Factor for snow (DDF snow). Summer balance (black bars) from point mass balance measurement at the Jungfraufirn (Fig. 1) and the corresponding DDF snow values in mm  $^{\circ}$ C<sup>-1</sup> day<sup>-1</sup> (red numbers), which are estimated during each year's summer mass-balance measurement period to determine a daily melt rate as an input parameter for the Community Firn Model (CFM).

# 3.4.2 Scaling precipitation data

The precipitation data from the Grimsel meteorological station at an elevation of 1952 m a.s.l. were scaled to an elevation of our study area of 3600 m a.s.l. The following method of Cullen and Conway (2015) was used to get the daily mean precipitation data at the Grosser Aletschgletscher from the Grimsel weather station. No precipitation data was recorded from the Grimsel weather station from 31 December 2011 till 30 November 2012. This data gap was filled by the mean winter daily precipitation data from 2004 to 2024. The daily rainfall data available at Grimsel can be linearly translated into daily SWE using the UBC

Watershed Model (Pipes and Quick, 1977) that employs the gentle variation in the rain and snow proportion as below.

for 
$$T \le 0.6$$
 °C : all precipitation is considered as snow for  $0.6$  °C 

**Figure 6.** The Figure illustrates the seasonal variability in the estimated SWE at two locations over the two decadal periods. Comparison of the SWE estimation during the accumulation period at the Grimsel Automatic Weather Station (red line) by translation of the recorded Snow Height (HS) to SWE using the Delta snow model, with the estimated SWE from seasonal point mass-balance measurements at the Grosser Aletschgletscher, Jungfraufirn (black line). Both curves show similar seasonal SWE variability with a mean bias of 1155 mm w.e., Pearson correlation of 0.6, and R<sup>2</sup> of 0.366.

#### 270 3.5 Internal Reflection Horizons as firn layers

To identify IRHs as annual firn layers, there must be more than one IRH on the radargram, and the distinguishable IRHs between the uppermost and lowermost IRH are also important. Within all three CMP data sets, we identified more than 25 IRHs (e.g., Fig. 4) in each data set. An iterative method was introduced to identify the picked IRHs as annual firn layers by estimating the SWE between the selected IRHs. This method uses the relationship of SWE with radar wave two-way travel time (twt), estimated interval velocity (Dix Eq. 1) from CMP semblance analysis, and density between the IRHs (Eq. 2). The velocity-depth and density-depth profiles obtained from the acquired CMP data at three accumulation areas of the glacier were used to estimate the SWE at the respective locations. The twt between identified IRHs within the acquired CMP profiles was subtracted and multiplied by the estimated layer velocity and density to obtain the SWE within each IRH as below.

$$SWE_{12} = \frac{twt_{12}}{2} \times V_{12} \times \rho_{12} \tag{7}$$

Here,  $\operatorname{twt}_{12} \times V_{12}$  is the thickness between  $IRH_1$  and  $IRH_2$ ,  $V_{12}$ : estimated layer velocity between two IRHs from CMP semblance analysis, and  $\rho_{12}$ : estimated density between two IRHs.

The SWE obtained between consecutive layers was compared chronologically against the estimated SWE from long-term point

measurements available at the Jungfraufirn location (Sake in Fig. 1) for the particular year. The SWE of the IRHs that matched the corresponding year's SWE values estimated from long-term point measurements was considered as the firn layer of that year. We believe that multiple reflection horizons may occur within a single firn layer due to a change in permittivity. Therefore, IRHs that did not fit the SWE from long-term point measurements were neglected as firn layers. This process was continued for all available IRHs from the CMP radargram. Using this approach, we identified 15 annual firn layers at Ewigschneefeld, 14 at the Mönchsjoch, and 10 at Junfraufirn locations (Fig. 1). In the case of the long GPR transect gathered at the Ewigschneefeld (Fig. 1), the mean twt of the IRHs along the profile was considered, and the difference in twt of each IRH was used to estimate the SWE of that layer as below.

$$twt_{12} = mean(twt_{IRH2}) - mean(twt_{IRH1})$$
(8)

Here in Eq. (8), the layer velocity and density were considered from the CMP gathered at the Ewigschneefeld (CMP3 in Fig. 1).

# 3.6 Spatial firn density and accumulation distribution



After identifying the internal reflection horizons (IRHs) as annual firn layers, we used the long GPR transect gathered at Ewigschneefeld (blue line in Fig. 1) to analyze the spatial variability in firn stratigraphy, density, and accumulation. The GPR profile, approximately 1.8 km long, spans an elevation range from approximately 3550 to 3380 m above sea level (Table 1). To understand how Alpine climatic conditions, such as precipitation and temperature, affect firn densification within this small elevation range, we estimated the spatial distribution of firn density and accumulation. This was done using the CMP-derived density-depth profile at the start of the GPR profile and a shallow density-depth profile (7.8 m) derived from direct measurements at the end of the profile (Fig. 1).

Figure 7. Illustration of the curve fit method to extend the glaciologically derived density-depth profile to deeper depths at the far end of the Ground Penetrating Radar (GPR) profile (Fig. 1). The density-depth trend derived from Common Mid-Point (CMP) data (black dots) and snow pit and firn core data (blue dots) at either end of the GPR profile gathered at Ewigschneefeld (Fig. 1) was fitted to the cubic polynomial of degree 3 (solid red line and black line, respectively). The density offset (grey fill) was estimated between interpolated CMP3 (orange line) up to a common depth of 7.8 m, and fitted density trends for the glaciological profile (black line). The shallow density-depth profile derived from the snow pit and firn core (blue dots) was extended to the deeper depth by assuming a cubic polynomial trend (dotted red line), which was shifted by the estimated average density offset. Later, these two (solid and dashed red lines) density profiles were used for spatial density distribution.

The density-depth trend from direct measurements, up to 7.8 m deep (Fig. 8), was fitted with a cubic polynomial (degree 3,  $R^2 = 0.93$ ) to identify the distribution pattern. We then compared the fitted curve with the interpolated CMP-derived density profile up to the same depth to find the average density offset (Fig. 7, orange line) by estimating the average difference between these two profiles. Further, the CMP-derived density trend was also fitted with a cubic polynomial (solid red line in Fig. 7), showing a good fit with the observations ( $R^2 = 0.75$ ). Due to the lack of density data beyond 7.8 m at the far end of the GPR profile, we assumed that the density trend continues following the cubic polynomial but with the estimated average density offset. The two density trends obtained at either end of the GPR profile were used to track the spatial distribution of firn density and accumulation. This was achieved by identifying annual layers and their corresponding depths using the previously



described method (Sect. 3.5) and interpolating the density distribution over the 1.8 km transect. The accumulation distribution along the GPR profile was then estimated by multiplying the interpolated density by the thickness of the identified firn layers (Eq. 7).

## 4 Results







## 4.1 Glaciological observations

We investigated the origin of the last summer horizon using direct observations from snow pits, snow cores, a firn core, and isotope analysis at two locations within the accumulation area of the Grosser Aletschgletscher. Figure 8 shows the density profiles obtained at Mönchsjoch and Ewigschneefeld from the snow pits, snow cores, and a firn core acquired during February-March 2024 (Table. 3 and Fig. 1). The snow cores obtained at these two locations extend to depths of 5.4 and 5.8 m, respectively, illustrating distinct density profiles with a noticeable density offset (nearly 100 kg m<sup>-3</sup>) in the upper few meters. However, at deeper depths (below 4 m), the density-depth trend is similar at both locations. In Figure 8, kinks can be observed in the density profile at depths of 2 and 3 m (Ewigschneefeld, SC1) and 3.5 m (Mönchsjoch, SC2), attributed to possible melt and refreezing events due to positive temperatures during early January 2024, late November, and mid-October 2023 (MeteoSchweiz, 2024), respectively. The data from the 4 m deep snow pit (SP1) next to the snow core (SC1) reveal a linear increase in the density profile without drastic changes (black line, Fig. 8). Further, the 3.8 m firn core (FC1) obtained from the bottom of the snow pit illustrates a similar increasing density trend, with a maximum density of around 640 kg m<sup>-3</sup> at a depth of 7.3 m. We identified ice lenses within the snow pit (SP1), snow cores (SC1 and SC2), and firn core (FC1) at both locations (Ewigschneefeld and Mönchsjoch), which are more frequent at deeper depths, hinting at the beginning of the last summer horizon and the possible effect of seasonal melt and refreezing. The observed data gap between the SP1 (red line) and FC1 (brown line) in Figure 8 can be attributed to measurement errors during the transition from the snow pit to the firn core sampling.

Isotope analysis supports the identification of the depth to the surface of the last summer horizon. The results in Figure 9 show the fluctuation of the  $\delta^{18}O$  and  $\delta D$  up to a depth of 4 m at Ewigschneefeld and beyond 5 m at Mönchsjoch (Fig. 1), illustrating the temperature variability during precipitation events within the 2023-2024 winter period. Below the 4 and 5 m depths, respectively at Ewigschneefeld and Mönchsjoch, there are no strong variations of isotopic signals, which can be interpreted as the dilution of the isotope signal due to summer melt. We therefore connect this change with the transition to the firn layer of the previous year (2023). With the available results from direct observations and isotope analysis, we assume the origin of the last summer horizon at an approximate depth of 4 m (Ewigschneefeld) and 5 m (Mönchsjoch) at two locations. The snow above the last summer horizon represents the winter precipitation until 29 February 2024 and equals 2000 mm w.e. at Mönchsjoch and 2100 mm w.e. at Ewigschneefeld.

Figure 10 represents the density profiles from the three shallow snow pits dug close to each CMP location to estimate the accumulation rate between 29 February and 17 May 2024. Density profiles obtained from snow pits at Ewigschneefeld (SP3) and Jungfraufirn near the long-term point measurements (SP4) show similar trends, with densities ranging from 200-460 kg m<sup>-3</sup> within 2 m of depth. However, due to unfavourable weather on 16 May 2024, only a 1.3 m deep snow pit was dug

**Figure 8.** Density-depth profiles obtained from direct measurements using glaciological methods such as snow-pit and firn cores at two locations (Fig. 1). The firn core (FC1) density profile is shown in brown solid line, and was drilled from the bottom of the snow-pit (SP1) depicted as a red line. The black line illustrates the density profile obtained from the snowcore (SC1) drilled a few meters parallel to the snow pit (SP1). All these measurements were taken at the Ewigschneefeld (Fig. 1). The blue line is the density profile estimated from the snowcore (SC2) drilled at the Mönchsjoch plateau (Fig. 1). The ice lenses observed at particular depths were denoted in corresponding coloured horizontal bars. The thickness of observed ice lenses is not to scale.

**Figure 9.** Comparison of  $\delta^{18}$ O and  $\delta D$  isotope samples gathered from the snow pit (SP1), snow core (SC2), and firn core (FC) at two locations (Ewigschneefeld and Mönchsjoch, Fig. 1). Blue and black dashed lines represent the beginning of the last summer horizon at Mönchsjoch and Ewigschneefeld, respectively.

at Mönchsjoch (SP2), presenting a density range of 240-360 kg m<sup>-3</sup>. The effect of temperature gradient due to elevation differences at Mönchsjoch, Ewigschneefeld, and Jungfraufirn is evident from the density profiles, exemplified by the number of ice lenses at SP3 and SP4 compared to SP2 locations (Fig. 10). It is also observable that ice lenses start at the near-surface level of around 0.25 m and depth hoar at 0.6-1.5 m, hinting at days with warmer temperatures between March and May 2024. The snow density does not exceed 360 kg m<sup>-3</sup> even at 1 m depth near Mönchsjoch (SP2). Whereas, at Ewigschneefeld (SP3) and Jungfraufirn (SP4), snow density reaches 430 kg m<sup>-3</sup> within 1 m depth, highlighting the spatial changes in snow compaction rate at different locations. Furthermore, the total SWE at SP2, SP3, and SP4 up to a common depth of 1.33 m is approximately 420, 510, and 480 mm, respectively. This variation in SWE across three locations can also be attributed to the spatial variation in density caused by melt and refreezing events between 29 February and 17 May 2024, which is further supported by the higher number of ice lenses at SP3 and SP4 (Fig. 10) compared to SP2.



**Figure 10.** Density-depth profiles obtained from the shallow snow pits dug near each of the three GPR CMP profiles (SP2-Mönchsjoch, SP3-Ewigschneefeld, and SP4-Jungfraufirn) during the May 2024 measurements. Dotted blue and dashed black horizontal lines show the Sahara dust layer and depth hoars only from the SP3. All ice lenses shown in light coloured horizontal bars were observed in the corresponding snow pit. The thickness of ice lenses, Sahara dust layers, and hoar is not to scale.

# 4.2 Ground Penetrating Radar-derived firn density profiles

Figure 11 summarises interval velocity-depth and density-depth results from the GPR CMP analysis at three accumulation areas of the Grosser Aletschgletscher. Hyperbolic reflections on the CMP profiles (e.g., Fig. 3) obtained at three locations (Fig. 1) represent the IRHs within the firn body. The CMP results are used to estimate the 1-D firn density-depth profile. The depth of the deepest discernible reflectors varies between the locations (Fig. 11) primarily due to changes in density with depth. Reflection patterns can be seen up to a maximum depth of approximately 37 m at Ewigschneefeld (CMP3), about 36 m at Mönchsjoch (CMP2), and a lower penetration depth of around 23 m at Jungfraufirn (CMP4, 1). The lower penetration depth at Jungfraufirn (CMP4 in Fig. 11) can be attributed to the high melt rate at this location resulting in stronger attenuation of the radar wave energy compared to the other two CMP locations. Meanwhile, the first pronounced IRH at the three CMP locations was identified at approximately 3 m depth. The radar wave velocity (interval velocity) and firn density fluctuate between 0.23-0.177 m ns<sup>-1</sup> and 320-850 kg m<sup>-3</sup> with depth and at three CMP locations, respectively.

The comparison of the results from the three CMP analyses highlights the increased density-depth trend at all locations, and the spatial variation in densification is also evident. At two locations (CMP2 and CMP3 in Fig. 1), densities range from 370-850 kg m<sup>-3</sup>, with noticeable kinks at approximate depths of 9, 12, and 16 m. The origin of these kinks is possibly due to changes in permittivity resulting from recent extreme melt and refreezing events in years such as 2022, 2021, and 2020 (Fig. 13). Further, we believe that the change in the density trend across three CMP locations is possibly due to years with relatively high precipitation and low melt (Fig. 13). The similar density-depth profile at the Jungfraufirn area (CMP4, red line in Fig. 11), with an initial density of approximately 320 kg m<sup>-3</sup> at about 3 m depth and reaches a maximum density of around 850 kg m<sup>-3</sup> at 22 m depth due to high melt and refreezing at this location. However, in the other two CMP locations, the density reaches its highest value (nearly 850 kg m<sup>-3</sup>) approximately at 30 m depths, implying spatial variability in firn densification.

# 4.3 Calibration of firn compaction models

The glaciological (direct) and geophysical (indirect, CMP) results provide 1-D firn density distribution data for several balance years. Here, we test firn compaction models against these results by forcing them with regional climate data from weather stations and tuning the model coefficients to understand how well the model output represents the observed geophysical and glaciological firn density profiles. The results of the uncalibrated KM and LIG models are illustrated in Figure 12 (a and b) at 3600 and 3400 m a.s.l., representing the Mönchsjoch and Ewigschneefeld locations, respectively. It is evident from the model results for Mönchsjoch that density increases with depth. At shallower depths (<9 m), the model density is well below the typical firn density value (550 kg m<sup>-3</sup>), and there is a shift in the density profile towards higher density below 10 m depth, reaching 550 kg m<sup>-3</sup> approximately 11 m depth. Beyond this point, the modelled density profile fluctuates and increases with depth, reaching a pore close-off density of 830 kg m<sup>-3</sup> approximately 50 m depth.

We ran the same models for a lower elevation (3400 m a.s.l.), the results are not significantly different, with the firn density (550 kg m $^{-3}$ ) reaching approximately 10 m depth and the pore close-off density (830 kg m $^{-3}$ ) at 50 m depth (Fig. 12b). The KM and LIG models exhibit indistinguishable variability at depths 

**Figure 11.** Interval velocity-depth (a) estimated from Vrms velocities picked from Semblance analysis using the Dix Eq. (1) and density-depth (b) profiles estimated from CRIM Eq. (2) using the Ground Penetrating Radar-based Common Mid-point Method (CMP) measurements (Fig. 1) at three locations of the Grosser Aletschgletscher. Sensitivity of interval velocity and density-depth profile for the Vrms picking in semblance analysis (Fig. 4) is shown in the shaded colours of the corresponding solid lines. More details on sensitivity analysis are discussed in section 5.3. All three CMP data sets were acquired on 16 and 17 May 2024.

results indicate a minor density offset between each other at the two elevations, despite changes in input parameters such as temperature and melt rate corresponding to the 200 m difference in elevation. It is identifiable that the modelled density profiles better represent the observational (CMP) results at Mönchsjoch (Fig. 12a) than at Ewigschneefeld and Jungfraufirn (Fig. 12b) in the untuned experimental setup. However, at shallower depths (<8 m), the model density trend is not comparable with the shallow glaciological density-depth profile, and both models underestimate the density with regard to the measurements despite the inclusion of the melt and refreezing processes within the LIG and KM models.



When the model coefficients were tuned (Table 3) to fit the observational results at both elevations (Figure 12c and d), the results better matched the point observational density profile by shifting the density profile towards higher values, reaching a firn density (550 kg m<sup>-3</sup>) at an approximate depths of 6-7 m (Fig. 12c) and 4-5 m (Fig. 12d), similar to the density trend obtained from CMP analysis at all three locations of the Grosser Aletschgletscher. Interestingly, after the tuning, the modelled

**Figure 12.** The tested Community Firn Model (CFM) density depth profiles for KM and LIG physics under the regional Alpine climate forcings to represent the observational results from snow pits, snow cores and a firn core (brown and black arrows) and GPR-based CMP-derived density depth profiles with sensitivity bars (black, blue and orange squares respectively at CMP2, CMP3 and CMP4) obtained during 2024 May measurements. (a) The KM (green) and LIG (blue) model results at Mönchsjoch, (b) Comparing the KM (red) and LIG (black) model at Ewigschneefeld density-depth results with the observations from CMP (blue and orange squares) and snow-pits (SP2, SP3 and SP4, black triangles), snow and firn cores (SC1, SC2 and FC, brown triangles). (c)The tuned KM and LIG model coefficients (green and blue squares, respectively) fit the observational results at Mönchsjoch (3600 m a.s.l). (d) illustrate the similar trend of density profiles from LIG and KM (black and red squares, respectively) at Ewigschneefeld (3400 m a.s.l). It can also be seen that the depth to firn density > 550 kg m<sup>-3</sup> (orange dashed line) and pore close-off density ( brown dashed line) for CFM and observed results.

density results exceed the pore close-off density (830 kg m<sup>-3</sup>) well before reaching 50 m of depth at Mönchsjoch (42 m) and Ewigschneefeld (40 m). At higher elevation (Fig. 12c), modelled density profiles do not depict the glaciological observations when compared to the untuned profile, but there is a slight shift in the density profile towards higher density. Whereas, at lower elevation (Fig. 12d), the model density trend depicts the fluctuations at shallower depths (<7 m), matching the glaciological density results. However, the KM and LIG models mimic the point density trend, primarily the CMP-derived, at both elevations better than the glaciological observations for the proposed tuning parameters (Table 3).

The sensitivity tests for three-degree day factors for snow values (3.5, 5.5, and 8.5 mm °C<sup>-1</sup> day<sup>-1</sup>) demonstrate similar density trends at both elevations for the LIG and KM models before and after tuning (Appendix Fig. A4). However, higher Degree Day snow (DDF snow) values shift the density profiles towards higher values even without model calibration, which illustrates the effect of increased melt on densification. The effect of the highest DDF snow value (8.5 mm °C<sup>-1</sup> day<sup>-1</sup>) is more pronounced at Ewigschneefeld compared to Mönchsjoch for the tuned LIG and KM models, with the depth to the pore close-off density reaching well below 40 m without calibration. The calibrated model results show increased density depth profile for higher DDF snow values (8.5 mm °C<sup>-1</sup> day<sup>-1</sup>, Fig. 12c and d), breaching the depth to pore closure of density at 37 and 34 m depths at Mönchsjoch and Ewigschneefeld, respectively.

## 4.4 Tracing the Accumulation history using Ground Penetrating Radar




The seasonal mass balance estimated from the manually measured snow depth and density from long-term point measurements at the Jungfraufirn accumulation area of the Grosser Aletschgletscher is illustrated in Figure 13. In the last two decades, the lowest winter accumulation was in 2021-22 (approximately 1150 mm w.e.), while the mean winter balance was around 1890 mm w.e., with a maximum of roughly 3025 mm w.e. in 2023-24. The SWE results indicate that the regional interannual precipitation variability fluctuates significantly over short periods. The summer of 2022 was a particularly extreme event within the last two decades, with peak ablation of approximately 2060 mm w.e., resulting in the complete loss of the annual firn layer and partial ablation of the 2021 summer layer (Fig. 13) at the Jungfraufirn location. Despite experiencing significant summer melt in the past eight years, the average summer mass balance over the last 20 years has remained around -450 mm water equivalent (w.e.). This average is maintained due to the occurrence of positive summer balances in specific years, such as 2016, 2014, 2013, 2007, and 2006. The annual SWE measurement (Fig. 13, grey bars) indicates the available yearly firn in mm w.e. after the melt season, which helps to interpret radar-derived SWE results to identify IRHs as firn layers, aiding in tracking the accumulation history.

Similarly, Figure 14 demonstrates the estimated SWE comparison between all three CMP measurements during the 16-17 May 2024 campaign and the GPR profile obtained at Ewigschneefeld on 29 February 2024, following the analysis in section 3.5. Figure 14 also shows the sensitivity of SWE estimation to the Vrms picking from semblance analysis, which is discussed in section 5.3. We compared all three CMP-derived SWE estimates with the SWE estimated from long-term point measurements to identify the IRHs as annual layers. The CMP gathered SWE near the Jungfraufirn (CMP4 in Fig. 1) agrees well with the estimated annual SWE from long-term point measurements ( $R^2 > 0.9$ ), as both were measured nearby. Additionally, the SWE estimated from CMP3 (at Ewigschneefeld) and CMP2 (at Mönchsjoch) correlates well with the estimated annual SWE from

**Figure 13.** The mass-balance of the Grosser Aletschgletscher glacier accumulation area from long-term point measurements near the Jungfraufirn area. The winter (blue) and summer balance (red) on the accumulation area were measured at the end of each season. The annual balance (grey) was estimated by adding the SWE from the winter and summer seasons of the last 20 years (GLAMOS, 2024).

long-term point measurements. However, there is an observable shift in CMP3-measured SWE at Ewigschneefeld and CMP2-derived SWE at Mönchsjoch for older firm layers, yet results are within an acceptable range ( $R^2 > 0.75$  for both locations). The GPR profile-derived SWE presents obvious variability across periods, which we attribute to uncertainties in considering the mean two-way traveltime (twt) for SWE estimation (discussed in section 5.4), and the corresponding R-square is lower for the GPR-derived SWE comparison ( $R^2 = 0.68$ ). Due to the strong melt event during the summer of 2022, despite the ablation of the entire layer from the previous season near the Jungfraufirn (Fig. 13), we expect some firn to remain at Mönchsjoch and Ewigschneefeld, which is discussed in section 5.4. The comparison of SWE indicates that a firn layer from the 2022-23 period, measuring approximately 600 mm w.e., remains present across the upper section of the Grosser Aletschgletscher. This finding is supported by the observed thinning of IRHs within the GPR transect radargram (Fig. 2). Additionally, Figure 14 (red square) reveals that no firn layer is present for the summer of 2022-23 at the CMP4 location. In contrast, the Mönchsjoch and Ewigschneefeld locations show the presence of the firn layer. Moreover, the 2021-22 firn layer is more ablated by approximately 920 mm w.e. at the Jungfraufirn location (Fig. 14, cyan bar), as compared to the CMP2 and CMP3 locations.



**Figure 14.** Illustration of identification of IRHs as annual layers and corresponding error analysis due to the uncertainties in internal Reflection Horizons (IRHs) picking in the CMP semblance analysis. Each error bar represents a Vrms picking uncertainty of 0.005 m ns<sup>-1</sup>, resulting in the 20-170 mm. w.e. fluctuation in the accumulation (Sect. 5.3). The plot also displays the corresponding R-squared value for fitting the SWE estimated from long-term point measurements and CMP-derived SWE for all GPR and CMP data collected during the 2024 expedition. The 2022 summer melt was extremely high, with complete ablation of the firn layer at the CMP4 (near Jungfraufirn 1) location and the further ablation of the 2021 firn layer (no red square in 2022 summer). The cyan bar in 2021 represents the remainder proportion of the 2021 firn layer after the 2022 summer melt season, whereas the blue bar is the measured 2024 winter accumulation obtained from long-term point mass-balance measurement. Due to the high melt and density at the CMP4 location, we identify firn layers up to 2015 (no red square after 2015). Similarly, maximum firn layers can be identified at the CMP3 location until 2010 (black square, but no blue square).

# 4.5 Spatial firn distribution



We traced the spatial distribution of firn density and accumulation along the GPR profile at Ewigschneefeld. Figure 15 demonstrates the spatial firn density within the identified firn layers along the GPR profile. As explained in Section 3.6, density estimates vary from higher elevations (approximately 3550 m a.s.l.) to lower elevations (3380 m a.s.l.) along the GPR profile. It is evident that firn density increases with depth along the transect (400-810 kg m<sup>-3</sup>). The influence of elevation change is noticeable, as we move to lower elevations (left to right in Fig. 15), and higher firn density is observed at shallower depths. For instance, the maximum firn density of approximately 780 kg m<sup>-3</sup> is seen at around 30 m (left), but the same density appears at a depth of approximately 21 m at the lower end of the GPR profile (Fig. 15, right side).

**Figure 15.** Spatial distribution of firn density along the GPR transect gathered at the Ewigschneefeld. Annual firn layers (black lines) are marked with the attributed years. X-axis is directed from the upper starting point (3550 m) to downvalley (end profile at 3380 m), along the GPR profile (Fig. 1).

Spatial variation in estimated SWE (Eq. 7) along the same GPR profile is illustrated in Figure 16, which represents the accumulation within identified annual layers. As the density within each layer varies due to overburden pressure, melting, and refreezing at various depths along the GPR profile, the estimated SWE within each layer changes due to variations in layer

thickness and density. This also demonstrates the spatial variability in the accumulation history, indicating that in the upper part of the GPR profile, the SWE is higher due to thicker firn layers, suggesting higher precipitation and lower melt. As we move downward along the profile, the thickness of the firn layer reduces, hinting at the possible lower accumulation. However, intense melting due to higher temperatures and related refreezing lead to denser firn layers. We can also observe that the 2024 winter precipitation above the last summer horizon (0-5 m depth) mostly remained constant along the profile (>1800 mm w.e), because of the early survey (winter 2024) the thickness was still rather constant, whereas later in the year the layer thickness would decrease due to the melt effect (as in deeper layers). However, as depth increases, the thickness of identified annual firn layers reduces with decreasing elevation along the profile (left to right in Fig. 16). Notably, thicker firn layers have higher accumulation (>4200 mm w.e) at deeper depths (e.g., depths between 22-28 m), attributed to denser and thicker firn layers as shown in Figure 15.



**Figure 16.** Spatial accumulation distribution within firn layers along the GPR transect gathered at the Ewigschneefeld. Accumulation varies within identified annual firn layers (black lines with corresponding years numbered in white) as the thickness and density vary along the GPR transect from higher to lower elevation (3550 m a.s.l on left to 3380 m a.s.l on right).

# 5 Discussion






## 465 5.1 Local firn density profiles using GPR

Our glaciological results provide the depth to the last summer horizon, as well as the firn structure and density profiles up to 7.8 m at the Ewigschneefeld and 5.4 m at the Mönchsjoch, covering a 2023 firn layer starting approximately at 4-5 m depth (Fig. 8). Identified cluster of ice lenses (depth < 4.8 m) within snow and firn cores hints at refrozen meltwater, complemented by the results from isotope analysis (Fig. 9). The glaciological results from both locations show a similar density structure except for the offset of nearly 100 kg m<sup>-3</sup> in the upper first meter. There is a possibility that the intense precipitation and wind gusts during the two-day gap between snow cores (SC1 and SC2) might have caused the surface snow density offset. Due to these weather conditions, data acquisition near the Jungfraufirn (Fig. 1) in March 2024 was not possible. Based on the combined measurements from SP1, FC1 (depth up to 5 m), and SP3 (Fig. 1), the estimated winter accumulation at Ewigschneefeld is approximately 2700 mm w.e. The in-situ measured accumulation at Jungfraufirn for the same period is around 3000 mm w.e.(2024 blue bar in Figs. 13 and 14), which results in an offset of about 300 mm w.e.. We assume this discrepancy arises for two reasons. First, we were unable to reach the surface of SP1 when we dug a shallow snow pit (SP3) on 17 May 2024. Second, the spatial variability between SP1, SP3, and SP4 at Jungfraufirn (Fig. 1), along with possible precipitation events between 17 May and 6 June 2024, likely contributed to the higher total winter accumulation.

The glaciological approach only provides point-scale results, which are limited to a shallow depth. For this reason, our investigation using geophysical means (GPR) helps to understand the firn structure and density profiles down to larger depths (>30 m). The application of GPR to derive firn density profiles is not a new approach. For instance, Brown et al. (2012) demonstrated the effectiveness of the GPR-derived CMP surveys in estimating density variations within the firn column at 13 locations along the Expédition Glaciologique Internationale au Groenland (EGIG) profile, covering the percolation area of the Greenland ice sheet (GrIS). However, to our knowledge, no studies are focusing on GPR-based CMP measurements to derive firn density estimations in Alpine glacier conditions. The results of the firn structure and 1-D firn density profiles derived from GPR-based CMP data provide a detailed understanding of the density distribution at different locations of the accumulation area of the Grosser Aletschgletscher (Fig. 11). Our investigation shows similar density profiles at Mönchsjoch and Ewigschneefeld (CMP2 and CMP3 in Fig. 11). The significant density changes at depths of 9, 12, 16, and 23 m are likely due to the intense summer ablation in 2022, 2021, 2020, and 2019, which had higher summer melt and lower winter precipitation, particularly in 2022 and 2020 (Fig. 13). The years with warmer summer conditions and relatively higher winter precipitation (years 2019, 2021, and 2023 in Fig. 13) likely resulted in high melting and refreezing processes, which might have caused the observed density variations at these depths.

The radar penetration depth varies at all three locations, indicating the attenuation of radar energy propagation. The significant attenuation, which reduces radar penetration to less than 23 m at CMP4 (Fig. 11), is possibly due to changes in the dielectric and conductivity properties of the subsurface (Davis and Annan, 1989). The study by Zhao et al. (2016) shows the increased radar wave attenuation attributed to the presence of liquid water, which is evident in temperate glaciers during summer. We believe the reduced penetration depth at the CMP4 location (Fig. 11) was caused by the effect of melt and refreezing at lower

elevations compared to the other CMP locations (CMP2 and CMP3 in Fig. 11). Similarly, the depth to pore close of density (830 kg m<sup>-3</sup>) at the CMP4 located at a lower elevation is around 20 m. Whereas, at high-elevation CMP2 and CMP3 locations (Table 1), the pore close-off density reaches nearly 30 m depth. With our spatial density-depth profiles across the different locations of the accumulation area, we reasonably conclude that processes such as surface melt, infiltration, and refreezing dominate the firn densification over the non-melt induced densification (including settling, sintering, and recrystallization).

## 5.2 Modelled firn densification







Many studies have developed firn compaction models and validated existing ones through direct physical observations, such as those by Herron and Langway (1980), Arthern et al. (2010), and Ligtenberg et al. (2011). As mentioned earlier, all these models were tested and calibrated for polar conditions using density profiles derived from firn core samples. However, studies like Huss (2013) used a modified version of the Herron and Langway (1980) firm densification model for temperate glaciers. The employed firn compaction model was calibrated using density measurements from 19 firn cores across various mountain glaciers and ice caps. Similarly, Sold et al. (2015) applied Reeh (2008b) firn layer compaction model, which is also based on the classical Herron and Langway (1980), assuming a linear relationship between firn densification and overburden pressure over time. This study calibrates the model with a scaling factor that depends on the measured and modelled IRH travel times. We demonstrate the use of LIG and KM models from the CFM modular run, forced with Alpine glacier regional climatic conditions, to primarily represent geophysical observations, by using the updated model coefficients (Table 3), to fit the CMP-derived density-depth trend (Sect 3.3). In Figure 12, we observed that both models behave similarly at both elevations, implying no noticeable impact from small variations in temperature and melt rate settings for 3.5 mm  $^{\circ}$ C<sup>-1</sup> day<sup>-1</sup> DDF snow. When the CFMs were not tuned, the model results showed an offset from the glaciological observations at lower depths (< 9 m), leading to an underestimation of shallow firn density. According to Ligtenberg et al. (2011), because of a lack of overburden pressure, densification rates at shallow depths are underestimated, and it is also suggested that discrepancy with modeled and glaciological density-depth profiles depended on the surface density assumption (300 kg m<sup>-3</sup>). So, any changes in the surface density assumption alters the beginning of the density-depth profile.

However, when we tuned the LIG and KM model coefficients (Table 3) to fit the CMP-derived firn density profiles, the offset at shallower depths reduced for the assumed surface density of 300 kg m<sup>-3</sup>. Our model results better fit the observed density-depth profiles after tuning (Fig. 12c and d), emphasizing the role of regional climatic conditions. According to Kuipers Munneke et al. (2015), these coefficients do not have a physical interpretation, which can be the same for different locations. However, coefficients could represent a process that is not included in these models. Any change in model coefficients can be linked to the changes in accumulation rates of the location, as the LIG and KM models are optimised for polar regions (Antarctica, Greenland). Therefore, a better model adaptation for Alpine conditions is needed, even without a direct relation to physical conditions. Similarly, we believe that our model results for the proposed coefficients, which align with the CMP-derived density-depth profiles (Fig. 12c and d), represent the accumulation rates of the Grosser Aletschgletscher; thus, the difference in our proposed coefficient (Table 3) compared to the LIG and KM coefficients can be reasonable. Calibrated model results illustrate stepwise changes in density at approximate depths such as 7, 9, 13, and 17 m (Fig. 12c and d). We argue

that these fluctuations in the density profile are due to the extreme summer melts in 2023, 2022, 2021, and 2020. However, these variations in modelled density structure are not as pronounced as in observed CMP density profiles. Further, both model results show that the depth to pore close-off density reaches beyond 45 m without tuning (Fig. 12a and b) and near 40 m after tuning the model coefficients (Fig. 12c and d). So the presented model results underestimates the CMP-derived densification at two elevations (CMP2 and CMP3 in Fig.1) where the depth to pore close-off density is around 30 m (Fig. 11). Thus, we interpret that the LIG and KM models, which are built for polar conditions need adjustment to use them in Alpine conditions so that these models can reflects the stronger firm densification rate that is predictable in Alpine conditions.

The sensitivity analysis shown in the appendix (Fig. A4d) indicates that we need to adjust the DDF snow value in order to reach the pore close-off depths determined by CMP analysis in Alpine conditions. This value is larger than the typical DDF snow value (3 mm °C<sup>-1</sup> day<sup>-1</sup>) used by studies for polar conditions (Wake and Marshall, 2015). However, the LIG and KM models' results show a slight difference in densification rate when we used the same DDF snow values for both elevations (Appendix Fig. A4a and c), with the depth to pore close-off reaching earlier at the lower elevation, hinting at the impact of lapse rate on firn densification for higher DDF snow values. The observed density variations for higher DDF snow values reflect the effect of extreme events; however, it is not evident enough as in the CMP-based density-depth profile. We investigated the dependence of the models on intra-annual precipitation variability while keeping the annual total precipitation constant (Appendix, Fig. A5). The results show that the timing of precipitation events within the year do not influence the vertical density distribution significantly, even though the deposition of precipitation occurs for different air temperatures. This indicates that the inter-annual thermal conditions are more important for densification than the temperatures during deposition (as long as precipitation occurs as snow and not rain).

In our study, we did not focus on the details of firn physical processes such as grain growth, water retention, and permeability. Instead, we aimed to test the models' ability to represent geophysically derived in situ firn densification results. However, we tried to interpret the LIG and KM model behaviour for two elevations (lapse rates) and three DDF snow sensitivity analyses. We demonstrate that our CFM results agree rather well with the CMP-derived firn density profiles when models are calibrated for CMP-derived density profiles and particularly match the depth to pore close-off density when tuned for higher DDF snow values. As we know, the physics of firn densification are not well represented except in laboratories. Even the firn compaction models in polar conditions are mainly semi-empirical, incorporating temperature and accumulation (Ligtenberg et al., 2011). On the other hand, we are able to show that these models can reproduce realistic density-depth distributions when combined with geophysical investigations. However, there is a need to improve the understanding of firn physics, mainly in Alpine conditions dominated by intense melt and refreezing, to obtain enhanced information on firn density and its evolution.

# 5.3 Sensitivity of field results

The GPR and CMP analyses not only provide the radar wave velocity profile within the firn but also help to derive the IRH depths. However, the semblance analysis used to extract the radar wave propagation velocity has uncertainties associated with picking the semblance wave peak, resulting in velocity measurement errors (Murray et al., 2007). According to Frolov and Macheret (1999), the velocity of radar waves depends solely on the relative permittivity of the medium. It is also quantified

that 3% water content would reduce the velocity by 0.03 m ns<sup>-1</sup> within the snowpack. Since we collected the CMP data at the end of winter when temperatures were well below freezing, we can ignore the effect of meltwater and assume a two-phase firn system (ice and air) for radar velocity, as explained in Bradford et al. (2009). However, the manual picking of the IRH and velocities is also affected by subjective errors.

Booth et al. (2011) study highlights the importance of precision in radar velocity estimation using Monte Carlo simulations. We followed a similar approach to understand the sensitivity in Vrms velocity picking from CMP semblance analyses. We explored two uncertainty values in Vrms picking before implementing the Dix Eq. (1) to get the interval velocity, which helps to derive the accurate firn physical properties. Our initial Vrms uncertainty value of 0.005 m ns<sup>-1</sup> falls within the suggested 50% contour (Booth et al., 2011) and the results of the SWE are shown in Figure 14. The resulting density fluctuation in all three CMP-derived density profiles falls in the range of 35 - 60 kg m-3 (Fig. 11). Further, we determine the error in SWE estimations for the identified annual layer to compare with the SWE estimated from long-term point measurements (Sect. 4.4) and Fig. 14). The fluctuations in SWE estimation for the identified annual layers using the interval velocity and density (Eqs. 1, 2, and 7) fall in the range of 20-170 mm w.e (Fig. 14) for all three CMP measurements.




Additionally, an IRH depth sensitivity of 1 ns with no change in Vrms picking could result in a ~45 mm w.e. variation in SWE estimation. The analyses show that an uncertainty of 1 ns in picking affects depth, radar velocity, and density estimation, which can cause SWE to vary between 40 and 75 mm w.e. However, this error is small compared to the uncertainty in Vrms picking. Further, we also assumed higher uncertainties in Vrms picking (0.01 m ns<sup>-1</sup>), which illustrates the increased interval velocity and density variations (~65 - 110 kg m<sup>-3</sup>), as well as the SWE in the range of 40-330 mm w.e. (Figure A6 in the Appendix). Whereas, for the GPR long transect, the estimated SWE from the identified firn layer is in the range of 25-140 mm w.e. The SWE error is amplified when we choose the mean twt for the identified firn layer from the GPR profile (orange cross in Fig. 14) obtained at the Ewigschneefeld. The identified firn layers in the long GPR profile show that the IRH thickness reduces as we move downhill along the profile (left to right in Fig. 2). Thus, the picked firn layer can vary by up to ~100 ns from one end to another (e.g., from ~186 ns to ~108 ns in Fig.2). As we select deeper firn layers, the variation in firn layer depth along the profile from one end to the other increases (e.g., from approximately 311 ns to 174 ns). Therefore, we propose to use the mean twt to estimate SWE along the GPR transect for the identification of the annual firn layer.

Our sensitivity analyses reveal that the estimation of interval velocity from the Dix Eq. (1) is sensitive to Vrms picking from the semblance analysis (e.g. Fig. 4). It is clear that the uncertainty in Vrms picking influences the interval velocity estimation (Dix Eq. 1), which in turn affects the firn density profile (Eq. 2) and resulting SWE required for identification of annual layers (14, Sect. 4.4). We also demonstrated the Monte-Carlo simulated results for different Vrms sensitivity values (0.001, 0.005, and 0.05 m ns<sup>-1</sup>), which show the higher variation in confidence interval of interval velocity for small variation in picked Vrms values (Apeendix Figs. A7, A8, and A9). So, we strongly support the use of the Monte Carlo simulation suggested by Booth et al. (2011) to provide the Vrms uncertainty range for density-depth profiles (Fig. 11) and the SWE estimations (Fig. 14).

## 5.4 Accumulation history and spatial firn distribution






In addition to the local firn density profiles at three locations in the accumulation area of the Grosser Aletschgletscher we also traced the accumulation history and spatial firn distribution (Figs. 14, 15, and 16). As stated earlier, we identified the many IRHs within the CMP data, and it should be noted that not all IRHs necessarily represent annual firn layers, because melt and refreezing events can generate high-density or ice layers, which can be misinterpreted as annual layers (Sold et al., 2015). Therefore, to identify the IRHs as annual layers, we introduced an iterative method (Sect. 3.5) by estimating the SWE between each IRH and validating it against the physically observed SWE from the long-term point measurements available at the Jungfraufirn (near the stake, Fig. 1). The CMP4 measurements are close to the long-term point observation, making it reliable to match the yearly SWE, which is evident from the high correlation ( $R^2 = 0.91$ ). The CMP2 and CMP3 comparisons show lower correlation ( $R^2 = 0.78$  and 0.79), which we attribute to the spatial variability in the firn distribution.

We assessed the role of extreme events, such as high melt during the summer of 2022, on the detection of firn layers before and after this event (2023 and 2021). As already explained (Sect. 4.4), the complete ablation of the 2022 seasonal layer (~2000 mm w.e) complicates the identification of firn layers from the IRHs detected at deeper depths. At the Jungfraufirn location, the 2022 firn layer does not exist (Fig. 13); however, this is not the case for the Mönchsjoch and Ewigschneefeld. This observation is supported by the radargram obtained from the GPR profile at Ewigschneefeld, demonstrating the strong IRH (at 140 ns in Fig. 2) that persists with a certain thickness in the upper part of the GPR transect and disappears as the profile reaches lower elevation. Furthermore, the Mönchsjoch (CMP2) and Ewigschneefeld (CMP3) are situated at higher elevations than CMP4 (Stake in Fig. 1 and Table 2), therefore we argue that the elevation difference (lapse rate) could have contributed to the survival of the 2022 firn layer at high elevated accumulation area (Sect. 4.4). To classify IRHs as annual firn layers, the iterative method helps in choosing IRHs chronologically and then estimating the SWE that fits the results from long-term point measurements (Fig. 14).

The spatial firn distribution presented in this study (Figs. 15 and 16) relies on CMP data to determine the average radar velocity above the reflector's depth. This approach assists in estimating the density and accumulation distribution across spatial scales. The spatial variability of accumulation within each layer is likely influenced by variable precipitation distribution linked to elevation differences and temperature lapse rates, causing significant melt at the lower end of the GPR profile. We believe that during extreme summer melt seasons, the meltwater percolates deeper until it reaches the firn, where the firn temperature is subzero for meltwater refreeze (in a cold firn glacier). Otherwise, the melted firn in summer percolates and drains through the impermeable horizon or crevasses of the glacier, thus resulting in mass loss. The deeper firn layers in the lower part of our GPR transect exhibit high accumulation (distance > 1000 m and depth > 20 m in Fig. 16). We attribute this to potential uncertainties in semi-automatic picking of IRHs using the phase-follower in ReflexW software due to a lack of visibility of IRHs at that location. To enhance the analysis, we recommend conducting more than one CMP measurement if the GPR profile indicates significant variations in firn stratigraphy along the transect.

## 5.5 Comparison with recent studies








A main difference between our study and that of Sold et al. (2015) at Findelengletscher is the usage of the GPR-based CMP method, which provides a solid estimation of depth to the IRHs, covering deeper firn layers and the local firn density profiles across the accumulation area of the glacier. The study by Sold et al. (2015) relied on the GPR travel time of each layer compared with the firn compaction model for density estimation. To translate the IRH travel time to depth, the Sold et al. (2015) study depends on the relationship between firn density and dielectric permittivity of firn (Frolov and Macheret, 1999) derived from firn cores and simple densification models, which also rely on repeat GPR measurements over consecutive years. In contrast, we extract the firn density-depth profile solely from the GPR-based CMP method without relying on firn core data and model results. However, misinterpretation of IRH affects the SWE estimation, which in turn alters the chronology of the annual firm layer by shifting the deeper layers. This issue was addressed in Sold et al. (2015) by comparing the layer SWE derived from GPR, firn core, and modelled hypothetical travel-time based SWE. Whereas, in our study, we rely on the SWE estimated from long-term point measurements to identify the IRHs as annual layers. The study of Sold et al. (2015) identified fewer (5-6) IRHs as firn layers, this is mainly due to the shallow firn cores and the use of a 400 MHz GPR antenna in temperate conditions, yielding a shallower radargram covering 5-6 annual layers (Fig. 2 in Sold et al. (2015)). Whereas in our study, the use of a high-frequency GPR antenna (500 MHz) in cold firn helps to penetrate deeper (> 300 ns) in firn (Fig. 2 in Sect. 3.1), and the iterative method of matching the GPR and CMP-derived SWE with the SWE from long-term point measurements helps to trace up to 15 annual layers (Fig. 14). However, CMP-derived accumulation history is constrained to point information, similar to firn cores and point mass-balance (in-situ) measurements. We addressed this issue by measuring the long GPR transect at Ewigschneefeld (Fig. 1) and identified more than 10 prominent IRHs down to a depth of more than 30 m (Figs. 2 and 15). The radargram shows distinctive and continuous firn stratigraphy along the profile. It should be noted that most IRHs run almost parallel to each other, and thickness reduces towards lower elevations (Figs. 2 and 15). This provides vital information about the spatial variability in the accumulation history and the melt rate. Comparing our GPR long transect with the Bannwart et al. (2024) study, in which the use of higher frequency GPR (800 MHz) reveals a shallower firn stratigraphy similar (Fig. 5c in Bannwart et al. (2024)) to that of our 600 MHz GPR radargram (Fig. 2). One can notice that both radargrams show similar thinning of firn stratigraphy with the reduced elevation (from left to right in radargrams). Here,a detailed discussion of Bannwart et al. (2024) GPR data is not the scope of our study, but we reserved the detailed comparison with our data presented in this paper, and recently gathered repeat measurements (winter 2025) as the subject of our future research on the temporal evolution of firn stratigraphy. However, we compared our glaciological density profile (SP1 and FC1 in Fig. 8) with the core location 3 density profile (Fig. 6a in Bannwart et al. (2024)), which shows a similar density trend as that of our snow pit and firncore density profile (Fig. 8), hinting at similar weather patterns between the two fieldworks.

The main drawback of our analysis with the firn layer identification from the GPR profile is the availability of just one CMP dataset (CMP3 in Fig. 1) on the uppermost part of the transect. The lack of CMP data at the lower part due to weather and time constraints forced us to assume the IRHs as straight lines to estimate SWE within each layer by considering the mean twt of each IRH (Sect. 5.3). The resulting firn layer SWE roughly correlates with the SWE estimated from long-term point

measurements ( $R^2 = 0.68$ ). The deeper firn layers (>200 ns in Fig. 2) show a dip in the IRHs and some discontinuities, which we speculate as effects of glacier dynamics causing the undulations. However, tracking the spatial firn distribution within the identified firn layers is possible by extrapolating the density-depth profile from 7.8 m to 36 m (Sect. 4.4) at the far end of the GPR transect. The effect of lapse rate due to elevation on firn densification (Fig. 7) is indicated by the shift in the extrapolated density profile compared to the CMP-derived density profile.

Despite some drawbacks, our study offers application of the GPR-based CMP method to derive firn density profiles, which is not time-consuming and non-invasive, unlike glaciological observations and modelling efforts in Sold et al. (2015), which also contain uncertainties associated with the radar travel-time to depth conversion. The introduction of an iterative approach in our study relies on SWE estimated from long-term point measurements to identify the IRHs as annual layers, offering reliability to track the accumulation history and spatial firn density. The combination of GPR (including CMP) and the firn compaction modelling approach provides the geometry and internal structure of the firn body. Our results show the potential to track the pore close-off depth (firn to ice transition) by tuning the firn compaction models with the CMP-derived density profiles. We believe that the non-invasive approach of GPR-derived density profiles helps calibrate the firn densification models, unlike other studies that rely on laborious in-situ glaciological observations. Our study demonstrates the importance of integrating in-situ geophysical, glaciological methods, and firn densification modelling to understand the evolution of firn structure and density. We believe such an approach is a significant step towards improving uncertainties in density assumptions in geodetically derived glacier volume-to-mass estimates, where firn density consideration is limited.

## 6 Conclusions







We introduced a novel approach to acquire firn structure, density, and accumulation history distribution on both local and spatial scales in the accumulation area of the Grosser Aletschgletscher using geophysical methods. Our approach relies on radar wave propagation velocity to accurately estimate the depths to the IRHs and trace firn density profiles in various parts of the accumulation area. Our new iterative method to identify IRHs as annual firn layers by chronologically estimating water equivalent within each IRH and comparing it with the SWE derived from long-term point measurements is different from the conventional approach of matching IRHs with firn core, or modeled firn density profiles, like in Sold et al. (2015). Use of a 500 and 600 MHz GPR antenna provided a compromise between radar penetration depth and resolution to identify enough IRHs (> 25 in all CMP data), which resulted in tracing the past 15 years' accumulation history. The long GPR transect provides clear evidence of ablation of 2022 firn due to extreme summer melt at lower elevation (below 3350 m a.s.l) and also helps to track the spatial firn distribution. Further comparing our GPR transect with the Bannwart et al. (2024) GPR profile shows a similar firn stratigraphy before 2021, which is promising for future studies on temporal firn densification.

Additionally, we proposed a new approach to calibrate firn compaction models using GPR-based CMP-derived firn density profiles under regional climate forcing rather than calibrating the firn densification models to firn core-derived firn density profiles. Our calibrated model results are promising, depicting a similar density trend as that of geophysical observations. This highlights the significance of these models in predicting firn density evolution in future studies in Alpine glacier conditions,

particularly in the absence of field data. However, we believe there is a need to adjust these models to depict the extreme summer melts that can be observed through geophysical and glaciological means. The application of established glaciological methods helped to identify the last summer horizon and thus support the interpretation of the indirect GPR measurements. Like other studies, our results are not exempt from limitations and uncertainties, particularly in interpreting the IRHs as annual levers. To address these uncertainties, we appear including at least one deep for each (darth > 15 m) equation more than two

Like other studies, our results are not exempt from limitations and uncertainties, particularly in interpreting the IRHs as annual layers. To address these uncertainties, we suggest including at least one deep firn core (depth >15 m) covering more than two annual layers near the CMP location in future studies for a direct comparison, which helps in the accurate interpretation of the IRHs as annual firn layers. We recommend using GPR during winter, especially in Alpine glacier conditions, as it provides deeper penetration depths and helps identify more firn layers. We are still exploring available firn compaction models in Alpine conditions, so we propose the consideration of other firn densification models in future studies, such as SNOWPACK (Bartelt and Lehning, 2002) and COSIPY (Sauter et al., 2020), which are commonly applicable in Alpine conditions, in combination with geophysical measurements to better address the extreme summer melt events that are common in the ongoing climate crisis, predominantly visible in Alpine conditions. Our study emphasizes the possibility of an intercomparison of firn structure, density, and accumulation history through geophysical, glaciological, and modelling approaches to mitigate the limitations of traditional firn core methods in challenging Alpine conditions. We believe the application of different methods proposed in this study helps to characterize the density contribution of the accumulation zones to improve the mean glacier density estimations.

**Figure A1.** The processed Common Mid-Point (CMP3) gather, which shows the pronounced Internal Reflection Horizons (IRHs) even at deeper depths (depth>250 ns or 25 m). Here, the x-axis is the distance of the CMP measurements, the left y-axis shows the radar two-way traveltime in ns, whereas the right y-axis is the radar-derived depth in meters with assumed radar velocity in firn as 0.21 m ns<sup>-1</sup>.

**Figure A2.** Illustration of the pickable Vrms velocity from the Semblance analysis of the CMP3 dataset for the deeper hyperbolas shown in Figure A1. It is possible to pick the hyperbolic reflections even up to 400 ns, and corresponding Vrms from the Semblance analysis (right figure). Here, the left y-axis is the radar two-way travel time in ns, whereas the right y-axis is in the depth domain in meters with assumed radar velocity in firn as  $0.21 \text{ m ns}^{-1}$ . The x axis shows distance in meters (left) and the pickable Vrms velocity in m ns<sup>-1</sup> (right).

**Figure A3.** Demonstration of picked Vrms (dotted lines) from all three CMP data sets (Fig. 1) and the corresponding estimated interval velocities (solid lines) using Dix Eq. (1).

**Figure A4.** Sensitivity analysis of the LIG and KM CFM models for different melt rates. Model's density profiles at the Mönchsjoch (a) and Ewigschneefeld (b) for the three different degree day factors of 3.5, 5.5 and 8.5 mm  $^{\circ}$ C<sup>-1</sup> day<sup>-1</sup> compared with the respective in-situ results (Fig. 1). The density profiles after tuning the LIG and KM models (c and d) at both locations to fit the observational results. Here, models were tuned for the same parameter coefficients but different DDF snow values.

**Figure A5.** Sensitivity analysis of the LIG and KM CFM models for the intra-annual precipitation variation, keeping the same annual accumulation. Case-1 shows the shuffling of the precipitation data by month, case-2 is by moving two months of precipitation data, case-3 is by shuffling precipitation within each month, and case-4 is the random shuffle of precipitation within the same year. Here, the shuffling of precipitation data is done only during the accumulation period (September-May).

**Figure A6.** Illustration of identification of IRHs as annual layers and corresponding error analysis due to the uncertainties in internal Reflection Horizons (IRHs) picking in the CMP semblance analysis. Each error bar represents a Vrms picking uncertainty of 0.01 m ns<sup>-1</sup>, resulting in the 40-300 mm. w.e. fluctuation in the accumulation (Sect. 5.3). The plot also displays the corresponding R-squared value for fitting the SWE estimated from long term point measurements and CMP-derived SWE for all GPR and CMP data collected during the 2024 expedition. The 2022 summer melt was extremely high, with complete ablation of the firn layer at the CMP4 (Jungfarujoch in 1) location and the further ablation of the 2021 firn layer (no red square in 2022 summer). The cyan bar in 2021 represents the remainder proportion of the 2021 firn layer after the 2022 summer melt season. The blue bar shows the measured 2024 winter accumulation. Due to the high melt and density at the CMP4 location, we identify firn layers up to 2015 (no red square after 2015). Similarly, maximum firn layers can be identified at the CMP3 location until 2010 (black square, but no blue square).

**Figure A7.** Interval velocity variation with depth for the Vrms sensitivity of 0.001 m ns<sup>-1</sup> implemented in Monte-Carlo simulation. The blue line is the mean interval velocity, and the gray shaded area is the fluctuation of interval velocity tested for the Vrms sensitivity (0.001 m ns<sup>-1</sup>). It is observable that the confidence interval spreads a lot as the depth increases for the tested Vrms value. The confidence interval also varies where the hyperbolic reflections are clustered at one place (near depth of 200 ns, Fig. A1).

**Figure A8.** Interval velocity variation with depth for a bit higher Vrms sensitivity value  $(0.005 \text{ m ns}^{-1})$  implemented in Monte-Carlo simulation. The blue line is the mean interval velocity, and the gray shaded area is the fluctuation of interval velocity tested for the Vrms sensitivity  $(0.005 \text{ m ns}^{-1})$ . It is observable that the confidence interval spreads a lot as the depth increases for small changes in Vrms values. At most of the depth intervals, the resulting interval velocity values reach beyond the physically possible values (grey area).

**Figure A9.** Interval velocity variation with depth for the Vrms sensitivity value of 0.05 m ns<sup>-1</sup> implemented in Monte-Carlo simulation. The blue line is the mean interval velocity, and the gray shaded area is the fluctuation of interval velocity tested for the Vrms sensitivity (0.05 m ns<sup>-1</sup>). It is observable that the confidence interval spreads beyond the physically possible interval velocity values for the glacier conditions, across all depth increases for small changes in Vrms values.

Data availability. The raw data are available at https://doi.org/10.5281/zenodo.17077546 (Patil et al., 2025). The meteorological data used for the firn densification modelling are available at https://www.meteoswiss.admin.ch/services-and-publications/service/open-data.html. The Grosser Aletschgletscher mass balance data are available at https://doi.org/10.18752/glrep\_series (GLAMOS, 2018).

Author contributions. A.P., with assistance from C.M., initiated and developed the study. A.P. wrote the paper and conducted the data analysis. The field campaigns were planned with contributions from C.M., T.S., and A.G. T.S. and A.G. analyzed the firn core, snow core, and snow pit data from the February-March 2024 campaign, while C.M. analyzed the snow pit data from the May 2024 fieldwork. Model run and analysis were done by A.P. A.B assisted in gathering the much-needed point mass-balance measurements and arranged the GPR equipment. All co-authors contributed to the reviewing and editing of the manuscript.

Competing interests. The authors declare that they have no conflict of interest.

Acknowledgements. The study was funded by the Bavarian State Ministry of Science and the Arts within the Elitenetwork Bavaria International Doctoral Programme "MOCCA - Measuring and Modelling Mountain Glaciers and Ice Caps in a Changing Climate". We would like to thank the following colleagues for their support during the expeditions in the accumulation area of the Grosser Aletschgletscher glacier in February-March 2024 and May 2024: Astrid Lambrecht, Felix Pfluger, Patricia Schlenk, Manuel Saigger, and Michael Stelzig. We appreciate Dr. C. Max Stevens from the University of Maryland for assisting us in understanding the application of the Community Firn Model (CFM) in our study. We also thank the International Foundation High Alpine Research Stations Jungfraujoch and Gornergrat (HFSJG), the Jungfraujochbahn, for the smooth conduct of our expeditions.

## **Financial support**

This research has been supported by the Elitenetzwerk Bayern (grant no. IDP M3OCCA) and the Deutsche Forschungsgemeinschaft (DFG) (grant number, SE 3091/5-1 awarded to Thorsten Seehaus).

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
