# Peer review of "Investigating firn structure and density in the accumulation area of the Grosser Aletschgletscher using Ground Penetrating Radar"

_EGUsphere, 2025_

## Referee Comment (RC1)

**Review of "*Investigating firn structure and density in the accumulation area of Aletsch Glacier using Ground Penetrating Radar*" - egusphere-2025-615**

Dear Authors,

I read your manuscript with interest – it's good to see links being made between properties derived from geophysical observations and climate models. Your approach parallels one that we applied for Norway's Hardangerjøkulen ice cap, and initially for Storglaciaren, hence I was intrigued to see further development of similar ideas and thus was keen to provide a review.

In general, I found your paper to be nicely presented and to provide good motivation for linking GPR and firn densification models. I fully agree that a suite of observational data are required if a reliable estimate of density is to be derived, including GPR velocity analyses ground-truthed against data from snow-pits and/or firn cores.

However, I am concerned that you may not have correctly evaluated your GPR velocities, and therefore that all quantities estimates and inferences derived from your GPR velocity model are likely in error. From what I understand of your reporting, you have taken velocities directly from your semblance picks (e.g., Figure 4 and Figure 11) and used them in Equation (1) to evaluate firn density. This would seem to be the case since, when Figures 4 and Figure 11 are overlaid (see below) the velocity plot and semblance picks appear to superimpose.

[Figure]

Figure 1. Comparison of velocity picks in semblance (authors' Figure 4) and velocity data ahead of density conversion (authors' Figure 11). The velocity trend in 'CMP2' looks to be very similar to that in the semblance picks.

The velocities that are picked from semblance analysis approximate "root mean square" velocity, $v_{RMS}$, a travel-time weighted average velocity of all velocities above a given reflection – i.e.,

$$v_{RMS_n} = \sqrt{\frac{\sum_{i=1}^{n} v_i^2 t_i}{\sum_{i=1}^{n} t_i}} \qquad (1)$$

where $v_i$ is the *interval* velocity through layer $i$, $t_i$ is the two-way transit time through layer $i$, and $n$ is a layer index. In order to establish any physical property, it is the interval velocity that is required from the velocity analysis, and it is the Dix Equation (Dix, 1955) that converts $v_{RMS}$ to $v_i$:

$$v_i = \sqrt{\frac{v_{RMS_i}^2 t_i - v_{RMS_{i-1}}^2 t_{i-1}}{t_i - t_{i-1}}} \qquad (2)$$

noting here that $t_i$ and $t_{i-1}$ are two-way travel-times to the $i^{th}$ and $i$-1$^{th}$ interfaces.

It may well be that you have implemented the Dix's Equation in your analysis and have just not reported it clearly in then main text, but the similarity between the above velocity trends suggests to me that this is not the case. For your $v_{RMS}$ model, which shows velocity progressively decreasing from 0.22 to 0.18 m/ns, converting to $v_i$ would give you a model that shows a more exaggerated velocity decrease with depth – and consequently, a more marked densification with depth.

In the figure below, I took ~25 $v_{RMS}$ points from Figure 4, assuming a decrease from 0.220 to 0.198 m/ns over travel-times from 60-140 ns, and expressed them as interval velocity. The divergence of these two trends is evident. If we apply the deepest $v_i$ in this model (~0.162 m/ns) in the CRIM equation used in the paper, and assuming $\rho_{ice}$ = 920 kgm$^{-3}$, the corresponding density is 1031 kgm$^{-3}$ (i.e., the interval velocity is lower than the reference value used for ice): your equivalent estimate is ~620 kgm$^{-3}$, which I indeed approximate if I use the deepest $v_{RMS}$ from the plot below.

[Figure]

Figure 2. Converting authors' picks of $v_{RMS}$ in Figure 4 to interval velocity. If I use (incorrectly) the deepest $v_{RMS}$ to evaluate density, I obtain something similar to the authors' values. If I use (correctly) the deepest interval velocity, the density is much higher.

Depending on which version of ReflexW is being used for analysis, the interval velocity model corresponding to any set of semblance picks is shown as a velocity model on the left-hand side

of the display. It is this model that you should use for parameter estimation, rather than that from the semblance picks themselves.

Unfortunately, until the GPR interval velocity (and thus density) is correctly evaluated, any correlation with products from climate models is invalid. Either that, or the manuscript must include detail of how interval velocity was actually defined.

I anticipate that you would want to correct their analysis and reconsider the match to climate models, and therefore I offer the following advice for any rewrite of the GPR analysis.

1. The presentation of the GPR data in Figures 3 and 4 could be improved. Although there is low signal-to-noise ratio at depth, it is almost impossible to see even the hyperbolae at shallower depth because picked reflection hyperbolae have been overlaid. I would consider plotting data with reflections unannotated for ease of comparison.
2. The semblance picks beyond ~200 ns depth lack credibility, especially when the corresponding reflection hyperbolae cannot be seen in the CMP gather.
3. A further refinement of semblance-based $v_{RMS}$ picks can be found here: this paper establishes that it is first-break travel-times which yield the most accurate physical properties, but only the higher amplitude half-cycles in the coda that yield semblance responses. The paper therefore presents a correctional methodology that the authors may wish to consider.
4. The authors appropriately consider the precision with which the semblance panel can be picked, suggesting a 0.005 m/ns picking error. It would be good to see this represented as error bars in the plots, and how it converts to density. I also think that this precision is greatly underestimated for the semblance picks with travel time >200 ns, and perhaps a larger uncertainty should be quoted for these.

I appreciate that this review will come as a disappointment, but I would encourage the authors to re-evaluate their velocity models and perhaps there may even be a better match to climate models.

Best regards,

Adam Booth

---

## Referee Comment (RC2)

**Review of manuscript** (https://doi.org/10.5194/egusphere-2025-615) by Akash M. Patil and colleagues entitled **"Investigating firn structure and density in the accumulation area of Aletsch Glacier using Ground Penetrating Radar"** submitted to The Cryosphere.

19 May 2025, Michael Zemp

**Summary**

Akash Patil and colleagues investigated the snow and firn structure of Grosser Aletschergletscher, Switzerland, based on snow pits, firn cores, Ground Penetrating Radar (GPR), and Common Mid-Point (CMP) measurements from field campaigns carried out between February and May 2024. They used these field observations and firn compaction models to reconstruct the accumulation history over the past 12 years and validated (or compared) their results against snow water equivalents from a long-term glaciological point mass-balance program run by Glacier Monitoring Switzerland.

**Evaluation**

Akash Patil and colleagues have conducted extensive fieldwork and investigated their observations in combination with process understanding, firn modelling, and existing measurements. This combination of methods provides new insights into the processes of snow/firn accumulation and densification in the accumulation zone of the largest Alpine glacier, which is changing from a dry zone to a percolation zone under current climatic conditions. The manuscript presents a valuable and very interesting study that fits well into the scope of The Cryosphere. Still, major revisions are required to improve the structure and readability of its text and figures, and to elaborate and discuss its key findings better. As such, it would be nice to add a direct comparison of their results from 2024 with an earlier survey from 2021 by Bannwart et al. (2024). My feedback is summarized as general and specific remarks outlined below.

**General remarks**

+ Structure: While the sections of the manuscript are generally well structured, the text would benefit a lot from better structuring, i.e., explaining the more general before going into details. This applies especially to the Abstract, Discussion, and Conclusions, as well as the captions for the Figures and Tables. I strongly recommend rewriting all captions, starting with (i) a short title, followed by (ii) a description of the content, (iii) explanations of axes, acronyms, and labels, (iv) notes or reading examples (if required), and (v) sources.

+ Readability: In its current form, the paper is hard to read, especially for "fast-food readers" who focus on the abstract, here-we-show-statement, figures, tables, and conclusions. To improve the readability, I encourage the authors to improve the structure and figure captions (see above), reduce the acronyms to the absolute minimum, and review the terminology (see below).

+ Terminology: The paper is not very consistent in its use of terms. Given the different Swiss place names (cf. Fig. 1), I suggest consistently using "Grosser Aletschgletscher". Write "Alpine" when

referring to the European Alps and "alpine" when referring to more general alpine environments. The (few essential) acronyms (e.g., GPR) should be introduced at their first use (only), and maybe written out in the captions. More questions related to terminology are listed in the specific remarks.

+ Key findings: Better emphasize the novelty and key findings of the study. From the Abstract and Conclusions, it is not clear (to me) which methods or combination thereof are novel and which findings are key for future research. Also, you mention a validation of your results against glaciological stake measurements in the abstract, while Figs. 6, 13, and 14 are instead a comparison.

+ Discussion: The discussion of the results remains relatively descriptive, and I see considerable potential for emphasizing the relevance of the findings. As such, you could discuss in more depth (and with the support of meteorological data) the presence (or absence) of seasonal and annual layers. Also, it would be interesting to discuss in more depth the impact of the increasing occurrence of summer melt in these formerly cold zones and the extreme summer 2022, which resulted in a net mass loss and erased an entire snow/firn layer. Finally, the authors mention that their GPR profile was a repeat measurement of the survey by Bannwart et al. (2024) from March 2021. It would be great to compare their GPR profiles (Bannwart et al. 2024, Fig. 5c) and firn cores (Bannwart et al. 2024, Fig. 6) from Ewigschneefeld, especially given the extreme years 2021/22 and 2022/23. Can you link the layers between the studies? Do you see any changes in the snow/firn densification profile? Figure 3 in Machguth et al. (2016) and Figure 4 in Sold et al. (2015) provide good examples of firn core comparisons.

The data from Bannwart et al. (2024) is temporarily available from Switch* and will later be added to the Zenodo repository related to the paper (https://10.0.20.161/zenodo.11071899).

*https://filesender.switch.ch/filesender2/?s=download&token=934552bd-6550-4398-a2ec-5837b0006f0e

+ Discussion & Figures: Your figures are all placed before the Discussion section. This is fine, but it would help to add (more) labels or reading examples that emphasize the topics discussed. As such, it would be interesting to see your age interpretation of the reflectors (in Figs. 2, 3, 4, 15, 16) and of seasonal/annual layers (Figs. 8, 9, 10, 11, 12).

+ Data availability: I strongly support publishing the dataset in a public repository such as Zenodo (https://zenodo.org/) or Pangaea (https://www.pangaea.de).

+ Methods: I note that GPR data processing and Common Mid-Point semblance analysis are outside my expertise. Here, I refer to the other reviewers and suggest considering their feedback on velocities picked from the semblance analysis (https://doi.org/10.5194/egusphere-2025-615-RC1).

**Specific remarks**

Title

L0: The present title is fine. However, if your firn modelling is a key finding of the study, you might consider reflecting this in the title.

Abstract

L1: Consider rewriting to improve structure and to emphasize key findings better.

L10: Remove line break in abstract.

Introduction

L18: Consider rewriting to improve the structure: background, state of the art, problem, "here we show…", general aim, and specific approach.

L35: Fix reference style of "Jordan et al., 2008".

L75ff: The study on Findelengletscher, by Sold et al. (2015), was mainly in temperate firn, while your study is located in cold firn. Does this difference matter when it comes to the consideration of radar velocity to get the internal reflection horizons? If so, please clarify.

L77-79: From the formulation of these lines, I would expect that your study will discuss the results compared to Huss (2013), and would not expect a comparison to Sold et al. (2015). I suggest rewriting the motivation part and better formulating the aim of your study.

Study area and data acquisition

L94: Concerning the largest glaciers, you might cite Windnagel et al. (2023).

L102: Instead of "validate", I would write "as available from GLAMOS (2024) and WGMS (2024)."

L116: Under the title "Glaciological investigations", I would also expect to find information about the point mass-balance measurements from GLAMOS that you used for validation/comparison to your results.

Methods

L158: Remove space in ($V_{firn}$).

L166: You have only one sub-section (3.3.1). Consider merging with Section 3.3. This would also reduce redundancies in the current title.

L177ff: Please provide more details on how the "tuning" was done.

L192: Would it be helpful to include these parameters in the sensitivity experiment (Section 5.3)? Also: replace ":" by ".".

L201: Check and adjust your term (e.g., "seasonal melt factor", "snowmelt rate value", "melt factor") to be consistent within your paper and, ideally, also with related key literature (e.g., Cogley

et al., 2011; Hock, 2003). Also, be careful when comparing point with glacier-wide, or seasonal to annual degree-day factors.

L208ff: The scaling of precipitation data includes major assumptions and comes with significant uncertainties. It would be good to add a corresponding discussion, and you might consider including it in your sensitivity experiment.

L224-225: Do you find a similar "trend" and/or "variability"? Please clarify.

L225: Figure 6 is a "comparison" rather than a "validation". Please clarify.

L228: Avoid acronyms in the title.

L229ff: Maybe something for the discussion concerning this Method section: How well are annual firn layers (or end of summer horizons) detectable at high-altitude sites? Possibility of complete melting of firn layers in extreme years 2022/23/24? Disturbance & mass transfer through strong melt events?

L262: Consider adding the $R^2$-value (0.88) to Figure 7.

Results

L285: "… to a depth of 5 m". From Figure 9, I would rather say at 4m at Ewigschneefeld (Site 1); for Mönchsjoch (Site 2), the fluctuations continue to the end around 5.5 m. Please clarify.

L292: "precipitation rate" or "accumulation rate"? Please clarify.

L295: What is the "effect of elevation difference"? Spatial variability or precipitation-elevation gradient? What about the temperature-elevation gradient?

L300ff: The difference between the SWE (420 mm vs 740 mm) is mainly due to the different depths of the snow pits, right? It might be better to compare the values at the common maximum depth. Please clarify.

L307-309: What influences the maximum depth of the reflection pattern? Mainly density? Please clarify.

L315ff: Could the change in density in Fig. 11 also originate from the percolating and refreezing meltwater from the intense summer melt of recent years?

L319: Instead of "testing and calibration", I would have expected a "calibration and validation" of the firn compaction model. Do you use these terms instead because you do not trust your glaciological and geophysical observations enough? Or is it related to missing uncertainties? Please clarify.

L340: Does the difference in density between model and glaciological observations indicate a process (e.g., refreezing) not included in the model? Please clarify.

L347: Avoid acronyms in titles. Write "accumulation" with a lowercase "a".

L350: Is "lowest winter precipitation" correct, or should it be "lowest winter accumulation"?

L355: The average of a summer balance that varies between accumulation and ablation might be misleading. Consider rewriting with a focus on the change from summer-accumulation to summer-ablation regime.

L370: Based on what evidence/observation/indication did you expect some remaining firn at Mönchsjoch and Ewigschneefeld? Please clarify.

L386ff: Can you provide more details on how you estimated SWE within each layer? Simply by the geometry of layers and density within layers? Does this estimate consider meltwater penetration through layers? Please clarify.

L388: "suggesting higher precipitation and lower melt": AND/OR?

L399: "…leading to lower accumulation as elevation decreases." I think lower annual net accumulation could also result from similar winter accumulation, but a less positive or negative summer balance with lower elevation. Consider and maybe rephrase.

L390: I think the uppermost layer (above the last summer horizon) is a special case since you only have winter accumulation but no summer balance (yet).

Discussion

L401: What are you referring to with "extreme weather" – "extreme" in a statistical sense, or just "extreme" with respect to "normal" lowland weather conditions? Please clarify.

L431ff: This section about firn densification modelling remains a bit fuzzy. Can you be clearer about lessons learned and conclusions for future use?

L447: From Fig. 12, I would not say that "the offset at shallower depths disappeared". Instead, I would say the offset could be reduced. Check and consider reformulation.

L457: "…the tested models consider …, liquid water percolation and refreezing…" What could be learned from the model on the effect of the extreme melt events of the past few years? Please clarify.

L456ff: The section on "Sensitivity of field results" seems valuable. However, I found your conclusions difficult to understand. Are there any other relevant parameters that should/could be checked? Or do you consider velocity picking to be the main uncertainty?

L465: Do you consider a velocity picking uncertainty of 0.005 m ns$^{-1}$ to be a typical or maximum uncertainty? How does Fig. 14 show the effect of an uncertainty of 1 ns? Please clarify.

L473-474: "Therefore, it is practical to use the mean TWT to estimate SWE along the GPR transect for the identification of the annual firn layer." Is this a justification for your approach or a conclusion for other studies? What is the corresponding state-of-the-art? Please clarify.

L475ff: The section on "Accumulation history and spatial firn distribution" is very interesting, but it is difficult to understand your main findings. As such, it would be interesting to be more specific about the advantage of your approach compared to that of Sold et al. (2015). Can you simulate the difference?

L475ff: The comparison to the earlier study (on the same GPR tracks) by Bannwart et al. (2024) is limited to one qualitative statement. It would be great to see a quantitative comparison of the results from both studies if feasible. Is it possible to link the GPR profile from the Ewigschneefeld? Do we see similarities or differences in the density profiles from the firn cores?

L478: "...it should be noted that not all IRHs necessarily represent annual firn layers." What else could they represent? Please clarify.

L484: "We assessed the role of extreme events...". How is your modelling affected by these extreme events? Do you expect some misinterpretation? Or do the models open the possibility for a detection & attribution of such events? Please clarify.

L491: "providing evidence for the survival of the 2022 firn layer." What evidence do you have? Please clarify.

L510: "The lack of CMP data at the lower part...". What was the reason of not having CMP data for Site 1?

Conclusions

L540ff: The Conclusions would benefit from rewriting, providing more structure, and highlighting key findings better: what was done;, what are the key results with respect to the accumulation history of Grosser Aletschgletscher, and from a methodological point of view; (what did we learn from a comparison to the earlier survey by Bannwart et al., 2024); what are recommendations for future work.

Appendix

L563, Fig. A1: I do not see a need for an appendix for one single figure and, hence, would rather integrate it into the paper.

L564ff: Data availability: For your own results, I strongly support publishing the dataset in a public repository such as Zenodo (https://zenodo.org/) or Pangaea (https://www.pangaea.de). For external input data, you can provide corresponding references (e.g., GLAMOS, MeteoSwiss).

L567ff: Author contributions: Who was responsible for the Community Firn Model runs and analysis?

Figures and Tables

All: Please improve the structure of the caption (see general comment).

Table 1: For the readability of the table, I would explain the acronyms (GRP, CMP) in the caption.

Fig. 1: Consider adding a note explaining why there is no CMP at Site 1. Regarding the background image, information on the platform, sensor, and date would be more relevant than the format. I suggest complementing Figure 1 with an additional Table summarizing the different observations per Site (1, 2, 3, 4), including locations, elevations, and survey dates.

Fig. 2: Consider adding the temporal interpretation of the internal reflection horizons (red lines). Indicate locations of other measurements (e.g., SP3, CMP3, SP1). It might be helpful to add a

comment on the elevation range from left to right of the profile and on prominent features, e.g., the merger of reflection horizons at a Distance of 1500m (melt of layers at lower elevation?) or the interpretation of the reflection horizon at a Time of 125ns between red lines.

Fig. 3: Consider adding the temporal interpretation of the internal reflection horizons (red lines). Add information on the location of this CMP(3?) with a reference to Fig. 1. Consider showing all CMPs in one figure for comparison.

Fig. 4: Explain all (colored) elements in figure, i.e. red lines (left), blue-green-yellow-red color range (right); xxx (right). Add information on the location of this CMP(3?) with a reference to Fig. 1.

Fig. 5: The caption does not seem consistent with the labels in the figures: "summer mass balance" (y-axis) versus "ablation" (caption)? "Degree-day" or "melt" factor? How does the width of the bars correspond to the (calendar or hydrological) years of the x-axis? Interestingly, there seems to be a correlation between summer balance and degree-day factor, maybe due to a feedback mechanism? What is the survey period of the summer balance? Was this consistent for all years?

Fig. 6: Provide information about the two graphs' trend, bias, and correlation.

Fig. 7: Consider adding a note on the density jumps at 3-5m, 12-17m, 23-27m.

Fig. 8: Consider adding the thickness of the ice lenses as values to the graph (e.g., next to the left y-axis). Consider adding a note on the density jump between the end of the red and the start of the brown line.

Fig. 9: Consider adding a note on the location of the end-of-summer horizons.

Fig. 10: Consider adding the data from Site 1 to this plot, too. Brown and orange horizontal bars are hard to differentiate. Consider adding the thickness of horizontal layers as values to the plot (e.g., right side).

Fig. 11: Avoid or explain the use of acronyms in the caption. Consider adding a note on the break in the velocity and density profiles at a depth of about 15 m.

Fig. 12: There seems to be an offset between snow pits/core and CMP values at common depths. Consider adding a corresponding note to the caption and/or cover in Discussion.

Fig. 13: Improve readability of the figure by separating bars of summer and annual balance. Add a horizontal line at zero balance. As a source, I would add a reference to GLAMOS (2024), and thank A. Bauder in the Acknowledgements.

Fig. 14: Does the x-axis provide calendar or hydrological years? Correct text to "43-72 mm w.e.". Consider adding a note explaining the lack of markers, e.g. red in 2021/22, red and orange before 2014. Add a horizontal line at zero balance. Consider labeling the years/dates of the estimated reflection horizons in Figs 2,3,4.

Fig. 15: Does the white at the bottom of the graph refer to ice (density >850 kg m-3) or the maximum GPR depth? Consider adding year/date labels to the identified firn layers.

Fig. 16: Is "Accumulation" the correct label for the color legend, or should it be "SWE"?

**References**

Bannwart, J., Piermattei, L., Dussaillant, I., Krieger, L., Floricioiu, D., Berthier, E., Roeoesli, C., Machguth, H., and Zemp, M.: Elevation bias due to penetration of spaceborne radar signal on Grosser Aletschgletscher, Switzerland, J. Glaciol., 1–15, https://doi.org/10.1017/jog.2024.37, 2024.

Cogley, J. G., Hock, R., Rasmussen, L. A., Arendt, A. A., Bauder, A., Braithwaite, R. J., Jansson, P., Kaser, G., Möller, M., Nicholson, L., and Zemp, M.: Glossary of glacier mass balance and related terms. IHP-VII Technical Documents in Hydrology No. 86, IACS Contribution No. 2, Organization, UNESCO-IHP, Paris, France, 2011.

GLAMOS: Swiss Glacier Mass Balance (release 2024), https://doi.org/10.18750/MASSBALANCE.2024.R2024, 2024.

Hock, R.: Temperature index melt modelling in mountain areas, Journal of Hydrology, 282, 104–115, https://doi.org/10.1016/S0022-1694(03)00257-9, 2003.

Huss, M.: Density assumptions for converting geodetic glacier volume change to mass change, The Cryosphere, 7, 877–887, https://doi.org/10.5194/tc-7-877-2013, 2013.

Machguth, H., MacFerrin, M., Van As, D., Box, J. E., Charalampidis, C., Colgan, W., Fausto, R. S., Meijer, H. A. J., Mosley-Thompson, E., and Van De Wal, R. S. W.: Greenland meltwater storage in firn limited by near-surface ice formation, Nature Clim Change, 6, 390–393, https://doi.org/10.1038/nclimate2899, 2016.

Sold, L., Huss, M., Eichler, A., Schwikowski, M., and Hoelzle, M.: Unlocking annual firn layer water equivalents from ground-penetrating radar data on an Alpine glacier, The Cryosphere, 9, 1075–1087, https://doi.org/10.5194/tc-9-1075-2015, 2015.

WGMS: Fluctuations of Glaciers Database, https://doi.org/10.5904/wgms-fog-2024-01, 2024.

Windnagel, A., Hock, R., Maussion, F., Paul, F., Rastner, P., Raup, B., and Zemp, M.: Which glaciers are the largest in the world?, J. Glaciol., 69, 301–310, https://doi.org/10.1017/jog.2022.61, 2023.

---

## Author Response (AR1)

Point by Point Response to the RC1 and RC2 comments on manuscript (https://doi.org/10.5194/egusphere-2025-615) by Akash M. Patil and colleagues entitled "Investigating firn structure and density in the accumulation area of the Grosser Aletschgletscher using Ground Penetrating Radar" submitted to The Cryosphere.

Within the revised manuscript, most of the introduction, discussion, and conclusion parts are restructured and rewritten as suggested by RC2. After considering the feedback from RC1, we updated the results section along with the Figs. 4, 7, 11, 12, 13, and 14 within the main text, and we added more figures to the appendix.

22.07.2025 by Patil et al

General comment from RC1: I am concerned that you may not have correctly evaluated your GPR velocities, and therefore that all quantities estimates and inferences derived from your GPR velocity model are likely in error. From what I understand of your reporting, you have taken velocities directly from your semblance picks (e.g., Figure 4 and Figure 11) and used them in Equation (1) to evaluate firn density. This would seem to be the case since, when Figures 4 and 11 are overlaid (see below), the velocity plot and semblance picks appear to superimpose.

**Response:**

We considered the suggestion of implementing the Dix equation to estimate the interval velocities within each reflection (IRHs) as recommended in the above description by RC1. Here we have attached updated figures representing the picked Vrms from the Semblance analysis (Fig. 1) and estimated Vi (interval velocity) through the Dix equation (Eq. 1). Figure 2 illustrates the application of the Dix equation to derive interval velocities from the corresponding semblance Vrms picks for all three CMP gathers.

$$V_{i} = \sqrt{\frac{V_{RMSi}^{2} twt_{i} - V_{RMSi-1}^{2} twt_{i-1}}{twt_{i} - twt_{i-1}}}.....1$$

Figure 1: Illustration of the CMP-based semblance analysis of the GPR data gathered at Mönchsjoch, Fig. 1. Here, it is observable that a radargram from the CMP gather (a), the corresponding coherence response of the IRHs on the CMP radargram (b), and 1-D velocity model with Vrms picks (dotted line) for the corresponding hyperbolae picks in figure (a) and interval velocity (solid line).

Figure 2: Presentation of velocity depth profile obtained from all three CMP gathers (Patil et al, 2025 TC Pre-print Fig. 1). The picked Vrms-depth profiles (transparent solid lines) are based on the semblance analysis and estimated interval velocity-depth profiles (Vi, solid thick lines) using the suggested Dix equation. Here, the y-axis represents the depth in two-way travel time.

1. The presentation of the GPR data in Figures 3 and 4 could be improved. Although there is low signal-to-noise ratio at depth, it is almost impossible to see even the hyperbolae at shallower depth because picked reflection hyperbolae have been overlaid. I would consider plotting data with reflections unannotated for ease of comparison.

**Response:**

We agree with the importance of improving figure representation. Figures 3 and 4 in the manuscript illustrate hyperbolic reflections picking using ReflexW software. We like to keep annotations that enhance the reader's understanding of CMP semblance picking.

Here we present an updated Figure 1, which replaces Figure 4 in the manuscript. This revised figure includes an additional figure (Fig. 1c) depicting the 1-D velocity model, linking it to hyperbolic reflections in the CMP radargram (Fig. 1a) and the corresponding coherence response observed in semblance (Fig. 1b).

2. The semblance picks beyond ~200 ns depth lack credibility, especially when the corresponding reflection hyperbolae cannot be seen in the CMP gather.

**Response:**

In our study, we deployed a 500 MHz Pulse-Ekko GPR system, as stated in the manuscript, to achieve high resolution while ensuring sufficient penetration depth in cold firn during the winter season. The primary advantage of using a 500 MHz GPR antenna is to balance the resolution and penetration depth. This is supported by the long GPR profile obtained using a 600 MHz IDS GPR system, which provides a penetration depth of up to 30 meters (Fig. 2).

While our CMP data exhibits decent resolution at deeper depths, and we can identify hyperbolae that correspond to semblance responses even at deeper depths (Figs. 3 and 1), which supports the credibility of our data.

Figure 3: The processed CMP data gathered at the CMP2 (Mönchsjoch, Fig. 1) location. It is noticeable that the presence of distinct hyperbolae extends to deeper depths (twt>350 ns). We can carefully distinguish hyperbolae even at deeper depths by using the plot scale option within the ReflexW software.

Figure 4 shows a zoomed-in version of processed CMP data (Fig. 1a & b) depicting the distinct hyperbolic reflections even at deeper depths (twt>300 ns). It illustrates pickable deeper hyperbolae for twt > 300 ns. We carefully identified hyperbolae (twt > 300 ns) by zooming in at specific depths. Hyperbolae at deeper depths are not easily visible without zooming in (e.g Fig. 1a), so we adjusted the plot scale accordingly, as shown in Figures 3 and 4, allowing us to pick them with precision.

Figure 4: Illustrating the zoomed version of Figure 1a & b that shows the pickable hyperbolae at deeper depths (twt>250 ns). It can be done using the plot scale option in the ReflexW software, which increases the gain of the GPR signal at deeper depths. Here, the left figure shows the annotated hyperboae on the CMP data, whereas the right figure shows the semblance Vrms pick corresponding to the hyperbolae.

4. The authors appropriately consider the precision with which the semblance panel can be picked, suggesting a 0.005 m/ns picking error. It would be good to see this represented as error bars in the plots, and how it converts to density. I also think that this precision is greatly underestimated for the semblance picks with travel time >200 ns, and perhaps a larger uncertainty should be quoted for these.

Response: We appreciate your suggestion regarding the depiction of error bars in density sensitivity, specifically for uncertainties of **0.005 m/ns** in Vrms and the corresponding changes in interval velocity (Vi). In Figure 5, we present the Vrms sensitivity results, which influence both interval velocity and the estimated density derived from the CRIM equation (Eqs. 1 & 2).

Figure 5: The GPR CMP-derived velocity-depth (a) and density-depth (b) profiles for all three CMP data obtained at the accumulation area of the Aletsch glacier (Patil et al, 2025 TC Pre-print Fig. 1). The effect of Vrms sensitivity (0.005 m/ns) while picking from the semblance analysis is also shown in both interval velocity and density depth profiles, as the shaded plots of the respective colour.

The selected Vrms uncertainty range of 0.005 m/ns accounts for the semblance contour (Booth et al, 2010) on both sides of the semblance response (Fig. 1 b) at the specific hyperbolic pick. To maintain consistency, we applied a constant Vrms sensitivity of 0.005 m/ns across all depths. This approach was chosen because sensitivity in Vrms picking increases with depth, and small variations lead to physically unrealistic interval velocity and density estimates.

3. A further refinement of semblance-based VRMS picks can be found <a href="here">here</a>: this paper establishes that it is first-break travel-times which yield the most accurate physical properties, but only the higher amplitude half-cycles in the coda that yield semblance responses. The paper therefore, presents a correctional methodology that the authors may wish to consider.

Response: We carefully considered your suggestion and implemented the Monte Carlo simulation to estimate interval velocities at three CMP locations. The following

results in Figure 6 (shown for CMP2 Vrms pick) illustrate the expected spread in the 95% confidence interval at deeper depths, where the resolution of the semblance response decreases with depth (Booth et al., 2010).

For this analysis, we assumed an uncertainty of 0.001 m/ns in Vrms picking from the semblance analysis (as seen in the figure for CMP2), and the estimated mean velocity from the Monte Carlo simulation reflects the corresponding interval velocity. Booth (2010) exemplified the sensitivity of interval velocity to the small variations in Vrms picks from the coherence response at deeper depths. So, we tested the impact of increased uncertainty in Vrms picking, which led to a broader spread in standard deviation, implying that variations in Vrms selection significantly influence interval velocity estimations that exceed the physically possible interval velocity range.

Figure 6: Monte-Carlo simulation to improve the accuracy of interval velocity estimation, which presents the firn physical properties (Booth et al, 2010). The blue line is the simulated mean velocity, which represents the interval velocity, and the red line is the estimated interval velocity from the Dix equation (Eq. 1). The shaded part shows the uncertainty range for a 95% confidence interval of the mean interval velocity.

**Response to RC2 General Remarks**

- + Structure: While the sections of the manuscript are generally well structured, the text would benefit a lot from better structuring, i.e., explaining the more general before going into details. This applies especially to the Abstract, Discussion, and Conclusions, as well as the captions for the Figures and Tables. I strongly recommend rewriting all captions, starting with (i) a short title, followed by (ii) a description of the content, (iii) explanations of axes, acronyms, and labels, (iv) notes or reading examples (if required), and (v) sources.
- + **Structure:** We appreciate your suggestions regarding the structure of the manuscript. We considered your remarks in revising the manuscript, along with a rewritten caption for all figures as suggested.
- + Readability: In its current form, the paper is hard to read, especially for "fast-food readers" who focus on the abstract, here-we-show-statement, figures, tables, and conclusions. To improve the readability, I encourage the authors to improve the structure and figure captions (see above), reduce the acronyms to the absolute minimum, and review the terminology (see below).
- + Readability: We considered your suggestions on readability of the manuscript, focusing on "fast food reader", we improved the structure, figures and table captions. We also reduced the number of acronyms; however, we want to retain the most commonly used and essential ones, such as GPR, CMP, CFM, KM, and LIG. We also reviewed and made the necessary changes in terminology.
- + Terminology: The paper is not very consistent in its use of terms. Given the different Swiss place names (cf. Fig. 1), I suggest consistently using "Grosser Aletschgletscher". Write "Alpine" when referring to the European Alps and "alpine" when referring to more general alpine environments. The (few essential) acronyms (e.g., GPR) should be introduced at their first use (only), and maybe written out in the captions. More questions related to terminology are listed in the specific remarks.
- **+ Terminology:** We agreed to the suggested changes to be consistent with terminology. We made the necessary changes in acronyms in the revised manuscript.
- **+ Key findings:** Better emphasize the novelty and key findings of the study. From the Abstract and Conclusions, it is not clear (to me) which methods or combination thereof are novel and which findings are key for future research. Also, you mention a validation of your results against glaciological stake measurements in the abstract, while Figs. 6, 13, and 14 are instead a comparison.
- **+ Key findings:** We made the suggested changes in presenting our work, primarily with a focus on scientific novelty in our research. Yes, we compared our geophysical results with

the stake measurements and also used the stake measurements in scaling the precipitation from Grimsel to Jungfraujoch (section 3.4.2 in the submitted manuscript). We changed the corresponding text while discussing the results of Figs. 6, 13 and 14.

- + Discussion: The discussion of the results remains relatively descriptive, and I see considerable potential for emphasizing the relevance of the findings. As such, you could discuss in more depth (and with the support of meteorological data) the presence (or absence) of seasonal and annual layers. Also, it would be interesting to discuss in more depth the impact of the increasing occurrence of summer melt in these formerly cold zones and the extreme summer 2022, which resulted in a net mass loss and erased an entire snow/firn layer. Finally, the authors mention that their GPR profile was a repeat measurement of the survey by Bannwart et al. (2024) from March 2021. It would be great to compare their GPR profiles (Bannwart et al. 2024, Fig. 5c) and firn cores (Bannwart et al. 2024, Fig. 6) from Ewigschneefeld, especially given the extreme years 2021/22 and 2022/23. Can you link the layers between the studies? Do you see any changes in the snow/firn densification profile? Figure 3 in Machguth et al. (2016) and Figure 4 in Sold et al. (2015) provide good examples of firn core comparisons.
- + Discussion: We appreciate your recommendation to emphasise our findings in detail. We considered your remark and tried to discuss the impact of the extreme summer of 2022. However, our firncore is not deep enough to depict the 2022 summer impact. We also discussed recent studies such as Sold et al. (2015), Machguth et al (2016), and Bannwart et al. (2024) by comparing them with our results in the revised manuscript. Our current manuscript aims at understanding the firn structure in detail using geophysical and glaciological measurements from the 2024 fieldwork, along with basic firn compaction modelling. Therefore, we argue that a quantitative comparison with Bannwart et al. (2024) is reaching too far at this stage. We already have an updated data set from the 2025 fieldwork (repeat measurements of the 2024 fieldwork), which we aim to use for analysing the temporal evolution of firn densification. In this context, we plan to carry out a detailed analysis and comparison of the existing data, including Bannwart et al. (2024) GPR measurements. However, in our revised manuscript, we compared and discussed the glaciological results from Bannwart et al. (2024).
- **+ Discussion & Figures:** Your figures are all placed before the Discussion section. This is fine, but it would help to add (more) labels or reading examples that emphasize the topics discussed. As such, it would be interesting to see your age interpretation of the reflectors (in Figs. 2, 3, 4, 15, 16) and of seasonal/annual layers (Figs. 8, 9, 10, 11, 12).
- **+ Discussion & Figures:** We introduced the labels on Figs. 15 and 16. We would like to keep the CMP dataset as simple as possible to help the reader understand the CMP semblance analysis (Figs. 3 and 4). We iteratively identified annual layers from Figs. 2, 3, 4 and 11 for which the labelling of annual layers makes all the mentioned figures more clumsy and chaotic. The same is applicable to Figs. 8, 9 and 10, in which we observe many ice lenses, and the introduction of ice lens thickness will make the figures unreadable. However, we provide more figures in the appendix corresponding to the specific figures in the main text. If at all needed, we can move them to the main text.

- **+ Data availability:** I strongly support publishing the dataset in a public repository such as Zenodo (https://zenodo.org/) or Pangaea (https://www.pangaea.de).
- **+ Data availability:** We are planning to publish the dataset in either Zenodo or Pangaea upon acceptance of the manuscript, as mentioned in the submitted manuscript, line 564.
- **+ Methods:** I note that GPR data processing and Common Mid-Point semblance analysis are outside my expertise. Here, I refer to the other reviewers and suggest considering their feedback on velocities picked from the semblance analysis (https://doi.org/10.5194/egusphere-2025-615-RC1).
- **+ Methods:** We responded to the RC1 comments with the updated results as in <a href="https://doi.org/10.5194/egusphere-2025-615-AC1">https://doi.org/10.5194/egusphere-2025-615-AC1</a>. Most of the results that depend on the CMP radar velocity picking have been changed, and the corresponding text has also been revised.

**Specific remarks**

**Title**

L0: The present title is fine. However, if your firn modelling is a key finding of the study, you might consider reflecting this in the title.

**Response:** We appreciate your suggestion; however, the present study primarily focuses on field methods to investigate the firn structure and density. Here, we aim to assess how well the firn compaction models represent the observations. We planned our future research on the detailed implementation of firn compaction models along with the geophysical observations. Therefore, we would like to keep the study title as it is now.

**Abstract**

L1: Consider rewriting to improve structure and to emphasize key findings better.

**Response:** Agreed to the suggestion. We improved the abstract as in the revised manuscript.

L10: Remove line break in abstract.

**Response:** Agreed, the line break has been removed.

**Introduction**

L18: Consider rewriting to improve the structure: background, state of the art, problem, "here we show...", general aim, and specific approach.

**Response:** We considered the suggestions; accordingly, the introduction has been revised as in the revised manuscript.

L35: Fix reference style of "Jordan et al., 2008".

**Response:** Thank you for pointing it out. We fixed the reference style.

L75<: The study on Findelengletscher, by Sold et al. (2015), was mainly in temperate firn, while your study is located in cold firn. Does this difference matter when it comes to the consideration of radar velocity to get the internal reflection horizons? If so, please clarify. Response: Yes, the temperate firn is characterised by the presence of liquid water, which can accelerate the densification process (Wakahama, 1975). Studies like Bradford et al (2009) reveal that even a small volume of liquid water content can alter the radar propagation velocity by more than 15%. Furthermore, because meltwater has a different permittivity than firn and ice, which could scatter electromagnetic waves and attenuate the radar signal deeper within the firn (Reinardy et al., 2019). Thus, the application of the GPR-based CMP method in temperate firn has a significant influence on radar propagation velocity and the depth estimation of the internal reflection horizons (IRHs). We discussed it in section 5.4 and also in the conclusion of the revised manuscript.

L77-79: From the formulation of these lines, I would expect that your study will discuss the results compared to Huss (2013), and would not expect a comparison to Sold et al. (2015). I suggest rewriting the motivation part and better formulating the aim of your study.

**Response:** Agreed to your suggestion. We reformulated the motivation part and rewrote the introduction as in the revised manuscript.

**Study area and data acquisition**

L94: Concerning the largest glaciers, you might cite Windnagel et al. (2023). **Response:** We appreciate your suggestion. We have considered it in the updated manuscript.

L102: Instead of "validate", I would write "as available from GLAMOS (2024) and WGMS (2024)."

**Response:** Agreed, we changed it as in the revised manuscript.

L116: Under the title "Glaciological investigations", I would also expect to find information about the point mass-balance measurements from GLAMOS that you used for validation/comparison to your results.

**Response:** Agreed, we used winter, summer and annual accumulation stake point measurements to compare our GPR-based CMP results as explained in the "Results" section of the submitted manuscript. We added more information within section 2.2, "Glaciological investigation", in the revised manuscript.

**Methods**

L158: Remove space in (Vfirn).

**Response:** Agreed, we changed it in the revised manuscript.

L166: You have only one sub-section (3.3.1). Consider merging with Section 3.3. This would also reduce redundancies in the current title.

Response: Appreciate your suggestions. We adapted the changes in the revised manuscript.

**L177<: Please provide more details on how the "tuning" was done.**

**Response:** We provided details of tuning as here "Parameter tuning was done by iteratively choosing the best coefficient that fits the GPR-derived CMP and glaciological observed density-depth profiles." We added this detail in the "Firn densification modelling" section 3.3 in the revised manuscript.

L192: Would it be helpful to include these parameters in the sensitivity experiment (Section 5.3)? Also: replace ":" by ".".

**Response:** We would not think so. We calibrated the model to fit the geophysical and glaciological observations. Table 3 shows the best-fitted parameter coefficients for the LIG and KM models' density-depth profiles to the observations. We believe that keeping the tuning parameters within section 3.3 is a better option rather than moving to the "Sensitivity..." section 5.3.

L201: Check and adjust your term (e.g., "seasonal melt factor", "snowmelt rate value", "melt factor") to be consistent within your paper and, ideally, also with related key literature (e.g., Cogley et al., 2011; Hock, 2003). Also, be careful when comparing point with glacier-wide, or seasonal to annual degree-day factors.

**Response:** Within the updated manuscript, we used the term Degree Day Factor for snow (DDF snow) as mentioned in Hock (2003). We also carefully considered the use of DDF snow in point and glacier-wide applications.

L208<: The scaling of precipitation data includes major assumptions and comes with significant uncertainties. It would be good to add a corresponding discussion, and you might consider including it in your sensitivity experiment.

**Response:** Agreed to your suggestions. We are working on the sensitivity of precipitation data and its influence on firn densification. We added the related discussion in Section 5.3, "Sensitivity analysis". Further, we discussed the results in the same section of the revised manuscript.

L224-225: Do you find a similar "trend" and/or "variability"? Please clarify.

**Response:** Yes, "variability" is the better choice of word. We changed the term "trend" to "variability" within the revised manuscript.

L225: Figure 6 is a "comparison" rather than a "validation". Please clarify.

**Response:** Agreed, we changed the term "validation" to "comparison" within the revised manuscript.

**L228: Avoid acronyms in the title.**

**Response:** Agreed to the suggestion. We have changed the acronyms in all titles within the revised manuscript.

L229<: Maybe something for the discussion concerning this Method section: How well are annual firn layers (or end of summer horizons) detectable at high-altitude sites? Possibility of complete melting of firn layers in extreme years 2022/23/24? Disturbance & mass transfer through strong melt events?

**Response:** Agreed, we discussed the suggested part within the "Discussion" section as below.

"At a high altitude site, direct observation, like isotope analysis (Fig. 9), helps to identify the last summer horizon, which can be supported by comparing GPR-based CMP estimated SWE with stake-derived SWE as explained in section 4.4 of the submitted manuscript". We also added more information regarding the extreme melts and the resulting mass transfer in the revised manuscript.

L262: Consider adding the R2-value (0.88) to Figure 7.

**Response:** Agreed, we updated Figure 7 with the R-squared values.

**Results**

L285: "... to a depth of 5 m". From Figure 9, I would rather say at 4m at Ewigschneefeld (Site 1); for Mönchsjoch (Site 2), the fluctuations continue to the end around 5.5 m. Please clarify. **Response:** Thank you for noticing a mistake. We corrected the mistake as pointed out and implemented it in the revised manuscript.

L292: "precipitation rate" or "accumulation rate"? Please clarify.

Response: It's an "accumulation rate".

L295: What is the "effect of elevation difference"? Spatial variability or precipitation-elevation gradient? What about the temperature-elevation gradient?

**Response:** In this case, we believe that the elevation difference is related to the temperature difference or a lapse rate. We changed the related text in the revised manuscript.

L300<: The difference between the SWE (420 mm vs 740 mm) is mainly due to the different depths of the snow pits, right? It might be better to compare the values at the common maximum depth. Please clarify.

**Response:** We appreciate your effort in spotting the mistake. Yes, here we compared the difference between the SWE for different depths, which should not be the case. The mistake was rectified, and we updated the text as in the revised manuscript.

L307-309: What influences the maximum depth of the reflection pattern? Mainly density? Please clarify.

**Response:** We mentioned the reason for the lower penetration depth in the following line within the same paragraph as "This lower penetration depth can be attributed to the high melt rate at this location compared to the other two CMP locations". We also discussed the variation in the radar penetration depth in section 5.1, lines 418-423 (submitted manuscript, https://doi.org/10.5194/egusphere-2025-615).

**L315<: Could the change in density in Fig. 11 also originate from the percolating and refreezing meltwater from the intense summer melt of recent years?**

**Response:** After RC1 comments, we updated most of the results (as in <a href="here">here</a>), including plots, and the CFM modelling part. We speculate changes in density at certain depths (approximately at 6, 8, 9, 12 and 16 m) due to intense summer melt. The depths of the identified annual layers support the speculation. We added a figure in the appendix explaining the same.

In the revised manuscript, we considered the required improvement in the "Discussion" section.

L319: Instead of "testing and calibration", I would have expected a "calibration and validation" of the firn compaction model. Do you use these terms instead because you do not trust your glaciological and geophysical observations enough? Or is it related to missing uncertainties? Please clarify.

**Response:** We agreed that the section title should be changed. The choice of the title had nothing to do with our measurements. The revised manuscript has a changed title for this section as "Calibration of firn compaction models".

L340: Does the difference in density between model and glaciological observations indicate a process (e.g., refreezing) not included in the model? Please clarify.

**Response:** The chosen semi-empirical LIG and KM models do consider the melting and refreezing processes to simulate the firn densification in polar climatic conditions. However,

KM and LIG models rely on a constant surface density assumption (in our case,  $300~kg~m^{-3}$ ) to reflect the density variation in shallow depth, as in glaciological observations. Surface snow density highly depends on temperature, precipitation, and wind speed, making it hard to model the shallow snow density to reflect observations (Ligtenberg et al., 2011). We considered your suggestions regarding the density difference between the model and glaciological observation and improved the text with further discussion in the same section of the revised manuscript.

L347: Avoid acronyms in titles. Write "accumulation" with a lowercase "a".

**Response:** Agreed, the updated manuscript has been corrected for similar mistakes.

L350: Is "lowest winter precipitation" correct, or should it be "lowest winter accumulation"? **Response:** We appreciate the mistake pointed out here regarding the terminology. It should be "lowest winter accumulation" instead of "lowest winter precipitation". We considered this change in the revised manuscript.

L355: The average of a summer balance that varies between accumulation and ablation might be misleading. Consider rewriting with a focus on the change from summer-accumulation to summer ablation regime.

**Response:** We agreed to rewrite the sentence as suggested. The revised manuscript has been corrected for the same.

L370: Based on what evidence/observation/indication did you expect some remaining firn at Mönchsjoch and Ewigschneefeld? Please clarify.

**Response:** We discussed this point in the section "Accumulation history and spatial firn distribution" line 487< as "At the Jungfraufirn location, the 2022 firn layer does not exist (Fig. 13), but this might not be true for Mönchsjoch and Ewigschneefeld. This argument is supported by the radargram obtained from the GPR profile at Ewigschneefeld, demonstrating the strong IRH (at 140 ns in Fig. 2) that persists with a certain thickness in the upper part of the GPR transect and the thickness reduces as the profile reaches lower elevation".

L386<: Can you provide more details on how you estimated SWE within each layer? Simply by the geometry of layers and density within layers? Does this estimate consider meltwater penetration through layers? Please clarify.

**Response:** Yes, the SWE estimation is straightforward, as you stated, by multiplying the geometry and density of the layers. We know the density from the CMP measurements and the estimated radar electromagnetic wave velocity within each identified layer. The SWE can be estimated using Eq.6 as in the manuscript. We believe that the estimated SWE within each layer considers the percolated meltwater that has refrozen at a particular depth of identified annual firn layers (Fig. 16 in submitted manuscript). However, we can not rule out the percolation of meltwater, which can not be quantifiable with this approach. We believe that percolated meltwater, which drains out of the glacier system, might not be accounted for within the identified layers.

We discussed these points in the "Discussion" section 5.4 of the revised manuscript.

**L388: "suggesting higher precipitation and lower melt": AND/OR?**

**Response:** The thicker firn layer at the upper part of the GPR profile (higher elevation) could be due to both "higher precipitation AND lower melt". Similar to the variation in the firn thickness at lower elevation due to "possible lower precipitation AND higher melt".

L399: "...leading to lower accumulation as elevation decreases." I think lower annual net accumulation could also result from similar winter accumulation, but a less positive or negative summer balance with lower elevation. Consider and maybe rephrase.

**Response:** We agree that the rephrasing would help in clarifying the statement. We changed the sentence in the updated manuscript.

L390: I think the uppermost layer (above the last summer horizon) is a special case since you only have winter accumulation but no summer balance (yet).

**Response:** Yes, it is indeed a special case, as our measurements cover up to the winter of 2024. We added this information in the revised manuscript.

**Discussion**

L401: What are you referring to with "extreme weather" – "extreme" in a statistical sense, or just "extreme" with respect to "normal" lowland weather conditions? Please clarify.

**Response:** Extreme in the sense of weather conditions like "intense precipitation and wind gusts". We changed the terminology within the revised manuscript to improve the clarity.

L431<: This section about firn densification modelling remains a bit fuzzy. Can you be clearer about lessons learned and conclusions for future use?

**Response:** We try to improve section 5.2 "Firn densification modelling", which includes the refined conclusion and future use as in the revised manuscript.

L447: From Fig. 12, I would not say that "the offset at shallower depths disappeared". Instead, I would say the offset could be reduced. Check and consider reformulation. **Response:** Agreed, we reformulated as in the revised manuscript.

L457: "...the tested models consider ..., liquid water percolation and refreezing..." What could be learned from the model on the effect of the extreme melt events of the past few years? Please clarify.

**Response:** Model results illustrate a sharp increase in density at depths such as 7, 9, 13 and 17 m (Fig. 12 c and d) after model calibration. We speculate that the sharp changes in density can be attributed to the extreme summer melts in 2023, 2022, 2021 and 2020 or change in density resulting from intense summer melts. However, LIG and KM models show some kinks at 6-10 m depth, but these fluctuations are not significant at deeper depths. We improved the CFM result interpretation in the revised manuscript after correcting for the RC1 comments, for which the CFM calibration of parameter coefficients is also changed (Table 3 in the revised manuscript).

L456<: The section on "Sensitivity of field results" seems valuable. However, I found your conclusions difficult to understand. Are there any other relevant parameters that should/could be checked? Or do you consider velocity picking to be the main uncertainty? **Response:** We believe that the main uncertainty lies with the picking of Vrms velocities from the CMP-based semblance analysis, which is illustrated by Booth et al. (2011). We also rewrote this section within the revised manuscript after implementing the comments from RC1.

L465: Do you consider a velocity picking uncertainty of 0.005 m ns-1 to be a typical or maximum uncertainty? How does Fig. 14 show the effect of an uncertainty of 1 ns? Please clarify.

**Response:** The choice of Velocity picking uncertainty of 0.005 m ns-1 is considered suitable because this uncertainty range falls within the 50% semblance contour (Booth et al, 2011). Uncertainty with 1 ns picking is very small when compared with the picking in Vrms sensitivity. So, the SWE and Density sensitivity bars or error bars (Fig. 14) already include the effect of 1 ns. We improved the text in the revised manuscript.

L473-474: "Therefore, it is practical to use the mean TWT to estimate SWE along the GPR transect for the identification of the annual firn layer." Is this a justification for your approach or a conclusion for other studies? What is the corresponding state-of-the-art? Please clarify.

**Response:** We propose the use of mean TWT to estimate SWE within each identified annual layer. It is a practical approach, as observed IRHs are significantly undulated, ranging from 311 to 174 ns. So, the mean TWT consideration for the further analysis is a better option. We updated the related text in the revised manuscript to improve the clarity.

L475<: The section on "Accumulation history and spatial firn distribution" is very interesting, but it is difficult to understand your main findings. As such, it would be interesting to be more specific about the advantage of your approach compared to that of Sold et al. (2015). Can you simulate the difference?

**Response:** The advantage of our method is to track the accumulation history without deep firn cores (depth > 20 m), and firn densification models to derive permittivity for IRH depth estimation. This can be possible using the GPR-based CMP method, which provides accurate IRH depth estimations (depth >30 m). The combination of CMP and long GPR transect helps to track the spatial accumulation and firn density distribution. We improved the section "Accumulation history and spatial firn distribution" in the revised manuscript, implementing suggestions to explain the differences between our study and Sold et al. (2015) in detail.

L475<: The comparison to the earlier study (on the same GPR tracks) by Bannwart et al. (2024) is limited to one qualitative statement. It would be great to see a quantitative comparison of the results from both studies if feasible. Is it possible to link the GPR profile from the Ewigschneefeld? Do we see similarities or differences in the density profiles from the firn cores?

**Response:** We thought about analysing the data from Bannwart et al. (2024); however, it does not suit the current aim of the manuscript. A new manuscript is under preparation focusing on the firn density evolution using temporal GPR data from Bannwart et al.(2024), Patil et al. (2025, TC Preprint) and our recently acquired winter 2025 repeat measurements. However, we compared the glaciological observations and discussed the results in the new subsection titled "Comparing recent studies" in the revised manuscript, which includes the Sold et al. (2015) and Bannwart et al. (2024) results comparison with our study.

L478: "...it should be noted that not all IRHs necessarily represent annual firn layers." What else could they represent? Please clarify.

**Response:** According to Sold et al. (2015), a large number of melt-refreezing events can generate high-density or ice layers. Therefore, IRHs can be identified as refrozen layers or ice lenses if there are no continuous GPR transects. Thus, we suggest the requirement of the chronological and iterative method for identifying IRHs as annual layers from GPR

CMP-based estimated SWE compared with stake-derived SWE. We discussed further in section 5.4 (submitted manuscript).

L484: "We assessed the role of extreme events...". How is your modelling affected by these extreme events? Do you expect some misinterpretation? Or do the models open the possibility for a detection & attribution of such events? Please clarify.

**Response:** Here, we can only speculate on the model interpretation concerning the Alpine climatic conditions, where the extreme events are quite common. Our current work is primarily focused on understanding how well firn compaction models represent field results. The detailed understanding of firn physics in the model results is reserved for our upcoming manuscript. However, we explained the interpretation of model results in Alpine conditions in the revised manuscript.

L491: "providing evidence for the survival of the 2022 firn layer." What evidence do you have? Please clarify.

**Response:** The presence of strong IRH (at 140 ns in Fig. 2) on the radargram obtained from the GPR profile at Ewigschneefeld, demonstrating the persistence of some firn layer of approximate accumulation of 500 mm w.e (Fig. 14) in the upper part of the GPR transect and the thickness reduces as the profile reaches lower elevation."

L510: "The lack of CMP data at the lower part...". What was the reason of not having CMP data for Site 1?

**Response:** Weather conditions and time constraints are reasons for not getting the CMP data at the lower part of the GPR transect.

**Conclusions**

L540

**Fig. 5:** The caption does not seem consistent with the labels in the figures: "summer mass balance" (y-axis) versus "ablation" (caption)? "Degree-day" or "melt" factor? How does the width of the bars correspond to the (calendar or hydrological) years of the x-axis? Interestingly, there seems to be a correlation between summer balance and degree-day factor, maybe due to a feedback mechanism? What is the survey period of the summer balance? Was this consistent for all years?

**Response:** The revised manuscript has an updated figure caption after considering the consistency in the label and the figure caption. The width of the bars has nothing to do with the years of the x-axis. Bar width has been chosen considering the plot visibility and to adjust the Degree-day factor for snow values on top of each bar. Yes with the increased DDF snow we have increased melting of snow which means more loss of snow in summer (as seen in the plot). Here we could expect the feedback mechanism.

The survey period is consistent between 1 May and 31 August.

Fig. 6: Provide information about the two graphs' trend, bias, and correlation.

**Response:** We agreed to provide the suggested information. The revised manuscript has an updated figure caption.

Fig. 7: Consider adding a note on the density jumps at 3-5m, 12-17m, 23-27m.

**Response:** We revised this figure after addressing RC1 comments. The updated figure has the suggested note in the caption.

**Fig. 8:** Consider adding the thickness of the ice lenses as values to the graph (e.g., next to the left y axis). Consider adding a note on the density jump between the end of the red and the start of the brown line.

**Response:** We noticed many ice lenses close to each other, mainly after 5 m depth. It makes it unreadable if we add thickness values in the plot. However, we provide an alternative figure in the Appendix with ice lens labels. We added a note regarding the density jump in the revised manuscript.

Fig. 9: Consider adding a note on the location of the end-of-summer horizons.

**Response:** We agreed to your suggestion. We added lines to explain the location of the last summer horizon in the revised manuscript.

**Fig. 10:** Consider adding the data from Site 1 to this plot, too. Brown and orange horizontal bars are hard to differentiate. Consider adding the thickness of horizontal layers as values to the plot (e.g., right side).

**Response:** We would like to keep this figure as it is now. We gathered this data between 16-17 May 2024, which has a significant time gap from the Site 1 data set acquired on 29 February 2024. The figure has many ice lenses or stratigraphy, adding a thickness axis makes it unreadable, as most of the ice lenses are close to each other in depth. However, we provided a labeled thickness figure in the appendix corresponding to the specific colour.

**Fig. 11:** Avoid or explain the use of acronyms in the caption. Consider adding a note on the break in the velocity and density profiles at a depth of about 15 m.

**Response:** The figure has changed after correcting for RC1 comments. We considered the required suggestions for this figure in the updated manuscript.

**Fig. 12:** There seems to be an offset between snow pits/core and CMP values at common depths. Consider adding a corresponding note to the caption and/or cover in Discussion. **Response:** We discussed the offset in the corresponding discussion section of the revised manuscript.

**Fig. 13:** Improve readability of the figure by separating bars of summer and annual balance. Add a horizontal line at zero balance. As a source, I would add a reference to GLAMOS (2024), and thank A. Bauder in the Acknowledgements.

**Response:** We agreed to the suggested changes, and the updated figure in the revised manuscript now has an added reference. Here, all winter and summer seasonal mass balance measurements date changed each year (GLAMOS, 2024). So, the figure has a constant winter, summer and annual survey date of the end of April, July and September, respectively, for better readability.

**Fig. 14:** Does the x-axis provide calendar or hydrological years? Correct text to "43-72 mm w.e.". Consider adding a note explaining the lack of markers, e.g. red in 2021/22, red and orange before 2014. Add a horizontal line at zero balance. Consider labeling the years/dates of the estimated reflection horizons in Figs 2,3,4.

**Response:** The x-axis provides measurement of summer mass balance date (GLAMOS 2024). We updated the figure with suggested changes of adding a note explaining the markers, and also added the line at zero balance. Adding years to Figures 2, 3 and 4 makes them chaotic. We would like to keep them as they are.

**Fig. 15:** Does the white at the bottom of the graph refer to ice (density >850 kg m-3) or the maximum GPR depth? Consider adding year/date labels to the identified firn layers.

**Response:** White space is because of no layers that can be picked beyond the maximum depth at a particular distance. It does not refer to the ice density. We agreed to add years to each identified firn layer. The figure is updated in the revised manuscript.

**Fig. 16:** Is "Accumulation" the correct label for the colour legend, or should it be "SWE"? **Response:** We thought about it, and we would like to keep it as an accumulation rather than SWE. Because our study aims at tracking spatial accumulation, which is nothing but the product of identified firn layers' thickness and the estimated layer density.

**References**

Bannwart, J., Piermattei, L., Dussaillant, I., Krieger, L., Floricioiu, D., Berthier, E., Roeoesli, C., Machguth, H., and Zemp, M.: Elevation bias due to penetration of spaceborne radar signal on Grosser Aletschgletscher, Switzerland, Journal of Glaciology, pp. 1–15, https://doi.org/10.1017/jog.2024.37, 2024.

Booth, A.D., Clark, R. and Murray, T. (2010), Semblance response to a ground-penetrating radar wavelet and resulting errors in velocity analysis. Near Surface Geophysics, 8: 235-246. https://doi.org/10.3997/1873-0604.2010008.

Bradford, J. H., Harper, J. T., and Brown, J.: Complex dielectric permittivity measurements from ground-penetrating radar data to estimate snow liquid water content in the pendular regime, Water Resources Research, 45, W08403, https://doi.org/10.1029/2008WR007341, 2009.

Brown, J., Bradford, J., Harper, J., Pfeffer, W. T., Humphrey, N., and Mosley-Thompson, E.: Georadar-derived estimates of firn density in the percolation zone, western Greenland ice sheet, Journal of Geophysical Research: Earth Surface, 117, 2011JF002089, https://doi.org/10.1029/2011JF002089, 2012.

Hock, R.: Temperature index melt modelling in mountain areas, Journal of Hydrology, 282, 104–115, https://doi.org/10.1016/S0022 1694(03)00257-9, 2003.

Ligtenberg, S. R. M., Helsen, M. M., and Van Den Broeke, M. R.: An improved semi-empirical model for the densification of Antarctic firn, The Cryosphere, 5, 809–819, https://doi.org/10.5194/tc-5-809-2011, 2011.

Machguth, H., MacFerrin, M., Van As, D., Box, J. E., Charalampidis, C., Colgan, W., Fausto, R. S., Meijer, H. A. J., Mosley-Thompson, E., and Van De Wal, R. S. W.: Greenland meltwater storage in firn limited by near-surface ice formation, Nature Clim Change, 6, 390–393, https://doi.org/10.1038/nclimate2899, 2016.

Patil, A. M., Mayer, C., Seehaus, T., and Groos, A. R.: Investigating firn structure and density in the accumulation area of the Grosser Aletschgletscher using Ground Penetrating Radar, EGUsphere [preprint], <a href="https://doi.org/10.5194/egusphere-2025-615">https://doi.org/10.5194/egusphere-2025-615</a>, 2025.

Reinardy, B. T. I., Booth, A. D., Hughes, A. L. C., Boston, C. M., Åkesson, H., Bakke, J., Nesje, A., Giesen, R. H., and Pearce, D. M.: Pervasive cold ice within a temperate glacier – implications for glacier thermal regimes, sediment transport and foreland geomorphology, The Cryosphere, 13, 827–843, https://doi.org/10.5194/tc-13-827-2019, 2019.

Sold, L., Huss, M., Eichler, A., Schwikowski, M., and Hoelzle, M.: Unlocking annual firn layer water equivalents from ground-penetrating radar data on an Alpine glacier, The Cryosphere, 9, 1075–1087, https://doi.org/10.5194/tc-9-1075-2015, 2015.

Wakahama, G., 1975. The role of meltwater in the densification processes of snow and firn. *International Association of Hydrological Sciences Publication*, *114*, pp.66-72.